# Local Discovery by Partitioning:
# Polynomial-Time Causal Discovery Around Exposure-Outcome Pairs

Jacqueline Maasch[1]    Weishen Pan[2]    Shantanu Gupta[3]    Volodymyr Kuleshov[1]    Kyra Gan[4]    Fei Wang[2]

[1]Department of Computer Science, Cornell Tech, New York, NY
[2]Department of Population Health Sciences, Weill Cornell Medicine, New York, NY
[3]Machine Learning Department, Carnegie Mellon University, Pittsburgh, PA
[4]Department of Operations Research and Information Engineering, Cornell Tech, New York, NY

## Abstract

Causal discovery is crucial for causal inference in observational studies, as it can enable the identification of *valid adjustment sets* (VAS) for unbiased effect estimation. However, global causal discovery is notoriously hard in the nonparametric setting, with exponential time and sample complexity in the worst case. To address this, we propose *local discovery by partitioning* (LDP): a local causal discovery method that is tailored for downstream inference tasks without requiring parametric and pretreatment assumptions. LDP is a constraint-based procedure that returns a VAS for an exposure-outcome pair under latent confounding, given sufficient conditions. The total number of independence tests performed is worst-case quadratic with respect to the cardinality of the variable set. Asymptotic theoretical guarantees are numerically validated on synthetic graphs. Adjustment sets from LDP yield less biased and more precise average treatment effect estimates than baseline discovery algorithms, with LDP outperforming on confounder recall, runtime, and test count for VAS discovery. Notably, LDP ran at least $1300\times$ faster than baselines on a benchmark.

## 1  INTRODUCTION

Uncertainty surrounding the true causal structure of observational data is a central challenge in causal inference. Unbiased causal effect estimation requires that certain variable types are omitted from covariate adjustment (e.g., colliders), while others are retained (e.g., confounders) [Schisterman et al., 2009, Lu et al., 2021, Holmberg and Andersen, 2022]. However, the identification of such variables can be challenging when the structural causal model is unknown, as is often true in practice. When domain knowledge is limited,

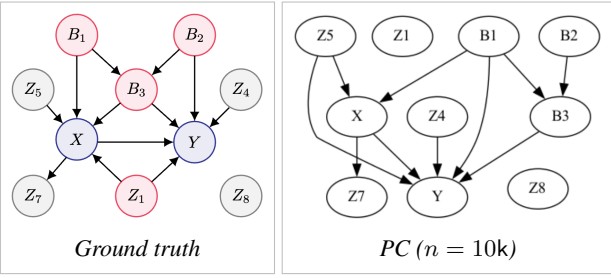

Figure 1: For sample sizes $n \leq 10$k, classic constraint-based algorithm PC fails to causally partition the data with respect to $\{X, Y\}$ (e.g., by misidentifying confounder $Z_1$ and instrument $Z_5$). Data generating process is linear-Gaussian (Fisher-z tests; $\alpha = 0.005$). See Figure A.1 for details.

causal discovery offers a powerful solution by automating the identification of critical variables and providing valid adjustment sets (VAS) for downstream inference.

We consider the set of causal discovery methods that do not require parametric assumptions on the data generating process for the identifiability of causal relations, increasing their reliability in real-world settings (e.g., healthcare). We therefore restrict our attention to constraint-based discovery, avoiding the standard parametric assumptions of functional causal models [Shimizu et al., 2006, Hoyer et al., 2008, Zhang and Hyvarinen, 2009, Rolland et al., 2022, Montagna et al., 2023] and restrictive assumptions on variable variances [Gao et al., 2020]. Despite asymptotic guarantees on correctness, the practicality of global constraint-based discovery is limited by the high sample and time complexity of running many conditional independence (CI) tests [Aliferis et al., 2010, Zhang et al., 2011, Schlüter, 2014, Zarebavani et al., 2020, Hagedorn et al., 2022, Braun et al., 2022]. As shown in Figures 1 and A.1–A.3, classic constraint-based methods like PC and FCI [Spirtes et al., 2000] can also display finite sample failure modes for VAS discovery, even with reasonably large sample sizes.

Increasingly, attention is shifting toward local discovery

methods that only learn relationships that are *causally relevant* to target variables of interest [Gupta et al., 2023, Cai et al., 2023, Dai et al., 2024]. However, existing local discovery methods that are tailored for VAS identification generally require that all variables are *pretreatment* (i.e., non-descendants of the exposure) [Entner et al., 2013, Cheng et al., 2023b]. This strong graphical assumption heavily simplifies the automated covariate selection problem, but is difficult to reliably verify in real-world data.

To address the challenges posed by existing global and local methods, we propose *local discovery by partitioning* (LDP): a local causal discovery algorithm designed for downstream inference tasks that does not assume pretreatment nor require parametric assumptions on the underlying data generating process. We approach this problem through the lens of *causal partitioning*, where variables are systematically subsetted according to their causal relation to an exposure-outcome pair. LDP returns a VAS under the backdoor criterion in worst-case polynomial time. Once a VAS is identified, conditional exchangeability holds, allowing the user to choose their preferred inference method to obtain unbiased effect estimates. In addition to VAS discovery, LDP identifies other variable types that can facilitate inference (e.g., instrumental variables [Imbens, 2014]) or statistical efficiency (e.g., causes of outcome [Brookhart et al., 2006]).

**Contributions** We introduce a taxonomy of eight exhaustive and mutually exclusive *causal partitions* that are universal properties of any arbitrary dataset with respect to an exposure-outcome pair. We then propose a polynomial-time procedure for leveraging these partitions to obtain a VAS under the backdoor criterion. LDP improves on the practicality of causal discovery in the context of downstream inference, owing to the following properties.

- *Time efficiency:* LDP only conducts tests that are needed for learning a VAS. The total number of independence tests performed is worst-case quadratic with respect to total variables, versus exponential for common baselines. On a community benchmark, LDP ran $1400\times$ to $2500\times$ faster than PC.
- *Sample efficiency:* The majority of CI tests defined in Algorithm 1 use conditioning sets of size one or two, contributing to more favorable sample efficiency relative to experimental baselines.
- *Flexibility:* LDP does not require parametric assumptions over the data generating process and does not assume the magnitude of the exposure-outcome effect (which may be null). We replace the pretreatment assumption with a milder, verifiable condition.

**Organization** The remainder of this paper is organized as follows. Section 2 describes preliminaries for causal graphical modeling, and Section 3 describes the universal partitioning taxonomy. Section 4 introduces LDP and establishes

that LDP returns a VAS for the true DAG under causal insufficiency, if a specific CI criterion is passed by at least one variable in the observed data. Section 5 compares LDP to existing works that do and do not assume pretreatment. In Section 6, we numerically evaluate LDP and establish that LDP achieves low runtimes and high sample efficiency when compared with existing causal discovery methods. Results demonstrate that VAS from LDP yield less biased and more precise average treatment effect estimates than baselines. Source code is available on GitHub.[1]

## 2 PRELIMINARIES

Univariate random variables are denoted by uppercase letters (e.g., $X$). Sets or multivariate random variables are denoted by bold uppercase (e.g., $\mathbf{Z}$), and graphs by calligraphic letters (e.g., $\mathcal{G}$). Let $X, Y, \mathbf{Z}$ denote continuous or discrete random variables representing an exposure, an outcome, and a variable set of unknown causal structure, respectively. Let $\mathcal{G}_{XY\mathbf{Z}}$ be the graph induced by $\{X, Y, \mathbf{Z}\}$. Sample sizes are denoted by $n$ and large values are abbreviated (e.g., $n = 1000 \rightarrow n = 1\mathrm{k}$.)

We restrict our attention to the set of causal graphs that are directed and acyclic.[2] We assume the common causal Markov condition and faithfulness [Spirtes et al., 2000].[3] We define active and inactive paths in $\mathcal{G}_{XY\mathbf{Z}}$ based on the concept of $d$-separation.[4]

**Definition 2.1** ($D$-separation, Spirtes et al. 2000)**.** Nodes $V$ and $V'$ in arbitrary causal DAG $\mathcal{G}$ are $d$-separated given node set $\mathbf{D}$ (where $\{V, V'\} \notin \mathbf{D}$) when there is no undirected path between $V$ and $V'$ that is *active* relative to $\mathbf{D}$.

**Definition 2.2** (Active paths, Spirtes et al. 2000)**.** An undirected path is considered *active* relative to a node set $\mathbf{D}$ if every node on this path is active relative to $\mathbf{D}$. A node $V \in \mathcal{G}$ is active on a path relative to $\mathbf{D}$ if

1. $V \notin \mathbf{D}$ is not a collider,
2. $V \in \mathbf{D}$ is a collider, or
3. $V \notin \mathbf{D}$ is a collider and at least one of its descendants is in $\mathbf{D}$.

We take an *inactive* path to be one that does not meet Definition 2.2 (e.g., due to existence of a collider $\notin \mathbf{D}$ on that path). As the definitions of active and inactive are with respect to $\mathbf{D}$, we assume $\mathbf{D} = \emptyset$ unless otherwise stated. We classify active paths between two nodes $\{Z, Z'\}$ following

---

[1] https://github.com/jmaasch/ldp

[2] Refer to Pearl [2009] for an introduction to graphical models.

[3] This ensures that the CI relations entailed by the joint distribution $p(X, Y, \mathbf{Z})$ precisely match those implied by the Markov condition applied to $\mathcal{G}_{XY\mathbf{Z}}$ (i.e., $p(X, Y, \mathbf{Z})$ and $\mathcal{G}_{XY\mathbf{Z}}$ are *faithful* to each other).

[4] Note that these definitions consider both the directed path and its corresponding undirected path, ignoring directionality.

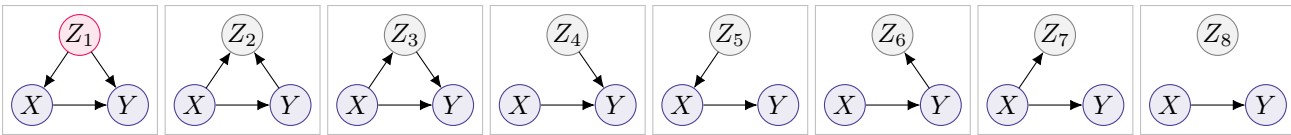

Figure 2: All potential acyclic triples that can be induced by $X$, $Y$, and a single $Z$ when paths are restricted to a length of 1.

from Table 2: 1) $Z \to \cdots \to Z'$, 2) $Z \leftarrow \cdots \leftarrow Z'$, or 3) $Z \leftarrow \cdots Z'' \cdots \to Z'$, where $Z''$ denotes a third node.

We say that causal association flows from exposure $X$ to outcome $Y$ through directed paths $X \to \cdots \to Y$. Non-causal association between $X$ and $Y$ due to a common cause also presents as statistical dependency, per Reichenbach's common cause principle [Peters et al., 2017]. Such common causes lie along *backdoor paths* for $\{X, Y\}$ ($X \leftarrow \cdots Z \cdots \to Y$).

**Definition 2.3** (Backdoor path, Pearl 2009)**.** Any non-causal path between exposure $X$ and outcome $Y$ with an edge pointing into $X$ ($\cdots \to X$).

We define valid adjustment with respect to the *backdoor criterion*, which enables *conditional ignorability* or *conditional exchangeability* for causal effect estimation in observational data: i.e., *confounding bias* is eliminated by achieving conditional independence between exposure $X$ and the potential outcomes of $Y$ [VanderWeele and Shpitser, 2013].

**Definition 2.4** (Valid adjustment under the backdoor criterion, Pearl 2009)**.** Let $\mathbf{A}_{XY}$ be an adjustment set for $\{X, Y\}$ that does not contain $\{X, Y\}$. $\mathbf{A}_{XY}$ is *valid* if

1. $\mathbf{A}_{XY}$ contains no descendants of $X$ and
2. $\mathbf{A}_{XY}$ blocks all backdoor paths for $X$ and $Y$.

**Definition 2.5** (Confounder, VanderWeele and Shpitser 2013)**.** A confounder for a variable pair $\{X, Y\}$ is a variable $Z$ for which there exists a variable set $\mathbf{S}$ (which may be empty) such that the effect of $X$ on $Y$ is unconfounded given $\{Z, \mathbf{S}\}$ but not given any proper subset of $\{Z, \mathbf{S}\}$.

## 3 CAUSAL PARTITIONS OF Z

We approach the problem of local discovery for downstream inference through the lens of *causal partitioning*, where variables are systematically subsetted according to their causal relation to the exposure and outcome. We establish an exhaustive taxonomy of eight disjoint partitions in Table 1. These partitions are not assumptions on the true DAG. Rather, they are *universal properties* of any ground truth DAG with respect to a chosen exposure-outcome pair. Thus, a true unique partitioning exists for any directed acyclic data generating process (though some partitions may be empty). In this work, we argue that these fundamental properties can be conveniently leveraged for efficient algorithm design.

In the next theorem, we establish that each variable $Z \in \mathbf{Z}$ belongs to a single ground truth partition.

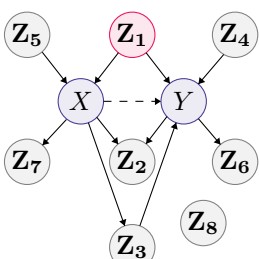

Figure 3: Given $X$ and $Y$, we can project any ground truth DAG onto a reduced 10-node DAG where nodes represent partition sets (which may be empty), arrows signify both adjacencies and indirect active paths (one or more), and inter-partition relations are abstracted away. The dashed edge suggests a possible null relation. Conditioning on $\mathbf{Z}_1$ blocks all backdoor paths for $\{X, Y\}$.

**Theorem 3.1.** *The eight partitions defined in Table 1 are exhaustive and mutually exclusive, such that any variable $Z$ falls uniquely under one partition category.*

*Intuition.* The intuition behind this taxonomy is reflected in the eight triple graphs in Figure 2, where $X$ is assumed to cause $Y$ and all paths are restricted to length one. These triples are exhaustive and mutually exclusive, and arise from simple enumeration of the three possible relations that one variable can take with respect to another: cause, effect, or neither. We generalize this intuition to the setting of arbitrary cardinality and indirect active paths, where the primitive relations of cause, effect, and neither map to the more complex relational combinations enumerated in Tables 2 and 3 (e.g., ancestor, non-ancestor, descendant, and non-descendant).

*Proof.* To prove Theorem 3.1, we define every type of active path from a candidate $Z \in \mathbf{Z}$ to $X$ or $Y$ that can possibly arise in the ground truth graph (Table 2). These can be direct adjacencies or indirect active paths of arbitrary length. Table 3 expresses every possible combination of path types that can coincide for a single $Z$. The mutual exclusivity of partitions follows from the fact that each cell of Table 3 contains a single partition, such that the pattern of allowable active path types from $Z$ to $X$ and $Y$ is unique for each partition. Exhaustivity follows from the fact that every cell in Table 3 that does not violate acyclicity contains a partition, such that all possible combinations are represented. $\square$

Some partitions coincide with existing terminology while others do not. $\mathbf{Z}_1$ approximately maps to *confounder* [Van-

$\mathbf{Z}_1$    *Confounders and their proxies*: Non-descendants of $X$ that lie on an active backdoor path between $X$ and $Y$ (Definition 2.5), and their proxies (Definition B.8).

$\mathbf{Z}_2$    *Colliders and their proxies*: Non-ancestors of $\{X, Y\}$ with at least one active path to $X$ not mediated by $Y$ and at least one active path to $Y$ not mediated by $X$.

$\mathbf{Z}_3$    *Mediators and their proxies*: Descendants of $X$ that are ancestors of $Y$, and their proxies (Definition B.8).

$\mathbf{Z}_4$    Non-descendants of $Y$ that are marginally dependent on $Y$ but marginally independent of $X$ (Definition B.3).

$\mathbf{Z}_5$    *Instruments and their proxies*: Non-descendants of $X$ whose causal effect on $Y$ is fully mediated by $X$, and that share no confounders with $Y$ (Definitions B.1 and B.8).

$\mathbf{Z}_6$    Descendants of $Y$ where all active paths shared with $X$ are mediated by $Y$.

$\mathbf{Z}_7$    Descendants of $X$ where all active paths shared with $Y$ are mediated by $X$.

$\mathbf{Z}_8$    All nodes that share no active paths with $X$ nor $Y$.

Table 1: Partitions are formally defined by the path combinations enumerated in Table 3.

| TYPE | ACTIVE PATH RELATIVE TO $X$ | ACTIVE PATH RELATIVE TO $Y$ |
|---|---|---|
| 1 | None (or none that do not pass through $Y$). | None (or none that do not pass through $X$). |
| 2 | $Z \to \cdots \to X$ path(s) and no other types. | $Z \to \cdots \to Y$ path(s) not passing through $X$ and no other types. |
| 3 | $X \to \cdots \to Z$ path(s) not passing through $Y$ and no other types. | $Y \to \cdots \to Z$ path(s) and no other types. |
| 4 | $Z \leftarrow \ldots Z' \cdots \to X$ path(s) and no other types. | $Z \leftarrow \ldots Z' \cdots \to Y$ path(s) and no other types. |
| 5 | Type 2 path(s) and Type 4 path(s). | Type 2 path(s) and Type 4 path(s). |
| 6 | Type 3 path(s) and Type 4 path(s). | Type 3 path(s) and Type 4 path(s). |

Table 2: Exhaustive enumeration of the types of active paths of arbitrary length that can lie between any variable $Z$ and $\{X, Y\}$. In confounded paths, $Z'$ denotes an additional variable in $\mathbf{Z}$ that may or may not belong to the same partition as $Z$. Note that some path types cannot coincide for a single $Z$, as they would induce a cycle.

| | | PATH TYPES RELATIVE TO $X$ | | | | | |
|---|---|---|---|---|---|---|---|
| | | TYPE 1 | TYPE 2 | TYPE 3 | TYPE 4 | TYPE 5 | TYPE 6 |
| RELATIVE TO $Y$ | TYPE 1 | $\mathbf{Z}_8$ | $\mathbf{Z}_5$ | $\mathbf{Z}_7$ | $\mathbf{Z}_5$ | $\mathbf{Z}_5$ | $\mathbf{Z}_7$ |
| | TYPE 2 | $\mathbf{Z}_4$ | $\mathbf{Z}_1$ | $\mathbf{Z}_3$ | $\mathbf{Z}_1$ | $\mathbf{Z}_1$ | $\mathbf{Z}_3$ |
| | TYPE 3 | $\mathbf{Z}_6$ | $\emptyset$ | $\mathbf{Z}_2$ | $\mathbf{Z}_2$ | $\emptyset$ | $\mathbf{Z}_2$ |
| | TYPE 4 | $\mathbf{Z}_4$ | $\mathbf{Z}_1$ | $\mathbf{Z}_2$ | $\mathbf{Z}_{2 \in \mathbf{M}_3}$ | $\mathbf{Z}_1$ | $\mathbf{Z}_2$ |
| | TYPE 5 | $\mathbf{Z}_4$ | $\mathbf{Z}_1$ | $\mathbf{Z}_3$ | $\mathbf{Z}_1$ | $\mathbf{Z}_{1 \in \mathbf{B}_3}$ | $\mathbf{Z}_3$ |
| | TYPE 6 | $\mathbf{Z}_6$ | $\emptyset$ | $\mathbf{Z}_2$ | $\mathbf{Z}_2$ | $\emptyset$ | $\mathbf{Z}_2$ |

Table 3: Permissible combinations of active path types relative to $X$ and $Y$. Cells contain partitions that can participate in the given combination. The empty set ($\emptyset$) indicates that this combination of active path types is forbidden under the acyclicity constraint. Subscript $\mathbf{M}_3$ denotes an M-collider, while subscript $\mathbf{B}_3$ denotes a butterfly-type confounder (Figure E.1).

derWeele and Shpitser, 2013], $\mathbf{Z}_2$ to *collider*, $\mathbf{Z}_3$ to *mediator*, $\mathbf{Z}_4$ to *pure prognostic variable* [Hahn and Herren, 2022], and $\mathbf{Z}_5$ to *instrumental variable* [Lousdal, 2018]. To our knowledge, $\{\mathbf{Z}_6, \mathbf{Z}_7, \mathbf{Z}_8\}$ do not coincide with existing terms in the causal inference literature. Further attention is given to defining $\mathbf{Z}_4$ and $\mathbf{Z}_5$ in Appendix B, given their role in the identifiability conditions of LDP (Section 4.1). Proxy variables are also further defined in Appendix B. When referring to multiple partitions collectively, e.g., $\mathbf{Z}_5$ and $\mathbf{Z}_7$, we use notation of the form $\mathbf{Z}_{5,7}$. When referring to a subpartition that is descended from or adjacent to a specific variable, we use notation of the form $\mathbf{Z}_{2 \in de(X)}$ and $\mathbf{Z}_{2 \in adj(X)}$, respectively.[5]

Within a single partition, there can be arbitrarily many active paths among its members (e.g., $Z_1 \to \cdots \to Z'_1$). Across partitions, active paths can exist in arbitrary DAGs as long as they comply with acyclicity and the patterns in Table 3.

**Definition 3.2** (Inter-partition active path). Any active path that is shared by at least two partitions, is *not* fully mediated by $X$ and/or $Y$, and complies with acyclicity and the combination of path types allowable in Table 3.

An example of an inter-partition active path that cannot exist

---

[5]An example of $\mathbf{Z}_{2 \notin de(X)}$ and $\mathbf{Z}_{2 \notin adj(X)}$ is $\mathbf{M}_3$ in Fig. E.1.

is $Z_4 \to \cdots \to Z_5$, as such a path violates the definitions of these partitions (Table 3, Appendix B). When we assemble all partitions into a single DAG, reduce active paths with $\{X, Y\}$ to length-1 arrows, and abstract away inter-partition active paths, we obtain the projection in Figure 3.

---

**Algorithm 1** *Local Discovery by Partitioning (LDP)*

---

**input** $X, Y, \mathbf{Z}$, independence test, significance level $\alpha$
**output** (1) VAS, if identifiable; (2) partition labels (as intermediate results)

1: Copy $\mathbf{Z}' \leftarrow \mathbf{Z}$
2: **for** all $Z \in \mathbf{Z}'$ **do**
   ▷ STEP 1: TEST FOR $\mathbf{Z}_8$
3:   **if** $X \perp\!\!\!\perp Z$ and $Y \perp\!\!\!\perp Z$ **then** $Z \in \mathbf{Z}_8$
   ▷ STEP 2: TEST FOR $\mathbf{Z}_4$
4:   **else if** $X \perp\!\!\!\perp Z$ and $X \not\perp\!\!\!\perp Z|Y$ **then** $Z \in \mathbf{Z}_4$
   ▷ STEP 3: TEST FOR $\mathbf{Z}_{5,7}$
5:   **else if** $Y \not\perp\!\!\!\perp Z$ and $Y \perp\!\!\!\perp Z|X$ **then** $Z \in \mathbf{Z}_{5,7}$
6: $\mathbf{Z}' \leftarrow \mathbf{Z}' \setminus \mathbf{Z}_4 \cup \mathbf{Z}_{5,7} \cup \mathbf{Z}_8$
   ▷ STEP 4: TEST FOR $\mathbf{Z}_{\text{POST}}$
7: **if** $|\mathbf{Z}_4| > 0$ **then**
8:   **for** all $Z \in \mathbf{Z}'$ **do**
9:     **if** $\exists\, Z_4 \in \mathbf{Z}_4: Z \not\perp\!\!\!\perp Z_4$ or $Z \perp\!\!\!\perp Z_4|X \cup Y$ **then**
      $Z \in \mathbf{Z}_{\text{POST}}$
10: $\mathbf{Z}' \leftarrow \mathbf{Z}' \setminus \mathbf{Z}_{\text{POST}}$
   ▷ STEP 5: TEST FOR $\mathbf{Z}_{\text{MIX}}$
11: **for** all $Z \in \mathbf{Z}'$ **do**
12:   **if** $Y \not\perp\!\!\!\perp Z$ and $Y \perp\!\!\!\perp Z|X \cup \mathbf{Z}' \setminus Z$ **then**
13:     $Z \in \mathbf{Z}_{1,2,3,5} \in \mathbf{Z}_{\text{MIX}}$
14: $\mathbf{Z}' \leftarrow \mathbf{Z}' \setminus \mathbf{Z}_{\text{MIX}}$
   ▷ STEP 6: SPLIT $\mathbf{Z}_{\text{MIX}}$ BETWEEN $\mathbf{Z}_{1,5}$, $\mathbf{Z}_7$, $\mathbf{Z}_{\text{POST}}$
15: $\mathbf{Z}_{\text{MIX}} \leftarrow \mathbf{Z}_{\text{MIX}} \cup \mathbf{Z}_{5,7}$
16: **if** $|\mathbf{Z}_{\text{MIX}}| > 0$ and $|\mathbf{Z}'| > 0$ **then**
17:   **for** all $Z \in \mathbf{Z}'$ **do**
18:     **if** $\exists\, Z_{\text{MIX}} \in \mathbf{Z}_{\text{MIX}}: Z_{\text{MIX}} \perp\!\!\!\perp Z$ and $Z_{\text{MIX}} \not\perp\!\!\!\perp Z|X$ **then**
19:       $Z \in \mathbf{Z}_1, Z_{\text{MIX}} \in \mathbf{Z}_{1,5} \notin \mathbf{Z}_{\text{MIX}}$
20:     **else** $Z \in \mathbf{Z}_{\text{POST}}$
21:   **for** all $Z_{\text{MIX}} \in \mathbf{Z}_{\text{MIX}}$ **do**
22:     **if** $\exists\, Z_{1,5} \in \mathbf{Z}_{1,5}: Z_{1,5} \perp\!\!\!\perp Z_{\text{MIX}}$ **then** $Z_{\text{MIX}} \in \mathbf{Z}_1$
23:     **else** $Z_{\text{MIX}} \in \mathbf{Z}_{\text{POST}}$
24: **if** $|\mathbf{Z}_{1,5} \cup \mathbf{Z}_1| > 0$ **then** $\mathbf{Z}_7 \leftarrow \mathbf{Z}_{5,7}$
   ▷ STEP 7: FINALIZE $\mathbf{Z}_1$ AND $\mathbf{Z}_5$
25: **if** $|\mathbf{Z}_{1,5}| > 0$ and $|\mathbf{Z}_1| > 0$ **then**
26:   **for** all $Z_{1,5} \in \mathbf{Z}_{1,5}$ **do**
27:     **if** $\exists\, Z_1 \in \mathbf{Z}_1: Z_{1,5} \not\perp\!\!\!\perp Z_1$ **then** $Z_{1,5} \in \mathbf{Z}_1$
28:     **else** $Z_{1,5} \in \mathbf{Z}_5$
   ▷ STEP 8: TEST FOR VAS ($\mathbf{Z}_5$ CRITERION)
29: **if** $\exists\, Z_5 \in \mathbf{Z}_5: Z_5 \perp\!\!\!\perp Y|X \cup \mathbf{Z}_1$ **then** VAS $\leftarrow \mathbf{Z}_1$
30: **else** VAS $\leftarrow$ {not identifiable}
31: {not identifiable} $\leftarrow \mathbf{Z} \notin \mathbf{Z}_1, \mathbf{Z}_4, \mathbf{Z}_5, \mathbf{Z}_7, \mathbf{Z}_8, \mathbf{Z}_{\text{POST}}$
32: **return** VAS and intermediate partition results

---

# 4 LOCAL DISCOVERY BY PARTITIONING

The pseudocode for LDP is expressed in Algorithm 1. Proofs of correctness are given in Appendix D.1. Given an exposure-outcome pair $\{X, Y\}$ and variable set $\mathbf{Z}$, LDP causally partitions $\mathbf{Z}$ in service of identifying a VAS under the backdoor criterion. LDP raises a warning if a VAS is not identifiable, as assessed using the $\mathbf{Z}_5$ criterion (Definition 4.3; Line 29 of Algorithm 1). A visual schematic for the learning process of Algorithm 1 is provided in Figure 4. Note that the correctness of certain intermediate results identified by LDP requires more stringent identifiability conditions than VAS discovery, as discussed in Remark 4.1.

*Remark* 4.1 (LDP is foremost a VAS discovery method, not a partition labeling method). LDP outputs partition labels as intermediate results en route to identifying a VAS. As intermediate results, LDP labels 1) five unique causal partitions ($\mathbf{Z}_1$, $\mathbf{Z}_4$, $\mathbf{Z}_5$, $\mathbf{Z}_7$, and $\mathbf{Z}_8$), and 2) a superset $\mathbf{Z}_{\text{POST}}$, which aggregates the remaining three post-treatment partitions ($\mathbf{Z}_2$, $\mathbf{Z}_3$, and $\mathbf{Z}_6$). While the sufficient conditions for guaranteeing correct partition labels for $\mathbf{Z}_4$, $\mathbf{Z}_7$, and $\mathbf{Z}_8$ are very lax (Theorem 4.6), sufficient conditions for correctly labeling the remaining partitions are significantly stronger than for VAS discovery (Section 4.1). For use of predicted partition labels *outside VAS selection*, we urge caution in interpreting the results. Thus, guaranteed partition label correctness for $\mathbf{Z}_1$, $\mathbf{Z}_2$, $\mathbf{Z}_3$, and $\mathbf{Z}_5$ in arbitrary DAGs is a key limitation and an area for future work.

**High-Level Overview** Here, we describe the basic logic of Algorithm 1 in plain English. Note that the partition labels assigned by Algorithm 1 assume that the sufficient conditions described in Section 4.1 are satisfied.

*Step 1* $\mathbf{Z}_8$ discovered with knowledge of $\{X, Y, Z\}$ only.
*Step 2* $\mathbf{Z}_4$ discovered with knowledge of $\{X, Y, Z\}$ only.
*Step 3* $\mathbf{Z}_7$ discovered with knowledge of $\{X, Y, Z\}$ only. $\mathbf{Z}_5$ will also be discovered if $|\mathbf{Z}_1| = 0$.
*Step 4* A fraction of $\mathbf{Z}_{\text{POST}}$ is discovered, providing complete knowledge of $\mathbf{Z}_6$ and partial knowledge of $\mathbf{Z}_2$ and $\mathbf{Z}_3$. This step leverages prior knowledge of $\mathbf{Z}_4$ that was obtained programmatically at Step 2.
*Step 5* $\mathbf{Z}_{\text{MIX}}$ is temporarily aggregated, providing partial knowledge of $\mathbf{Z}_1$, $\mathbf{Z}_2$, $\mathbf{Z}_3$, and $\mathbf{Z}_5$. $\mathbf{Z}_{\text{MIX}}$ is a transient superset that is used to differentiate $\mathbf{Z}_1$ and $\mathbf{Z}_5$ from $\mathbf{Z}_{\text{POST}}$ in Step 6.
*Step 6* Knowledge of $\mathbf{Z}_{\text{POST}}$ is complete. $\mathbf{Z}_{\text{MIX}}$ is fully disaggregated, providing final partition labels for some members and moving others to superset $\mathbf{Z}_{1,5}$. At this stage, we also finalize our knowledge of $\mathbf{Z}_7$. By Line 19, all members of $\mathbf{Z}_5$ have been placed in $\mathbf{Z}_{1,5}$. By Line 22, members of $\mathbf{Z}_1$ that are adjacent to $Y$ have been uniquely identified.
*Step 7* $\mathbf{Z}_1$ and $\mathbf{Z}_5$ are fully disentangled. This step tests whether a member of superset $\mathbf{Z}_{1,5}$ is marginally

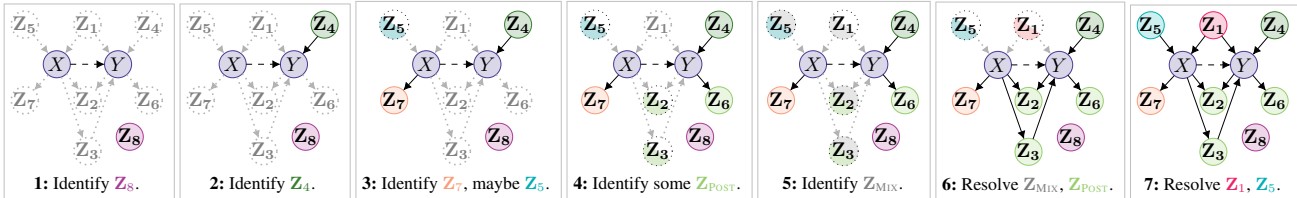

Figure 4: Each step of Algorithm 1 reveals additional information about the partitions of $\mathbf{Z}$ without requiring LDP to learn the full causal graph. Nodes that are fully colored are fully discovered, partial coloring denotes partial knowledge, and no coloring denotes no knowledge.

dependent on known members of $\mathbf{Z}_1$. All previously known members of $\mathbf{Z}_1$ are adjacent to $Y$. $\mathbf{Z}_1$ that are left to be discovered are those with indirect active paths to $Y$. In any arbitrary DAG, no $Z_5 \in \mathbf{Z}_5$ will ever be dependent on a $Z_1 \in \mathbf{Z}_1$ that is adjacent to $Y$. However, all $\mathbf{Z}_1$ are marginally dependent on at least one $Z_1 \in \mathbf{Z}_1$ adjacent to $Y$.

*Step 8* The algorithm concludes by testing the $\mathbf{Z}_5$ criterion, which indicates whether a VAS was discovered and raises a warning when failed.

**Time Complexity** We report Big $O$ complexity in terms of total independence tests performed, as is conventional for constraint-based causal discovery [Spirtes et al., 2000, Tsamardinos et al., 2006]. The first for-loop (Steps 1–3) requires a linear number of tests in $O(|\mathbf{Z}|)$, where Step 1 caches all marginal test results for every candidate relative to $\{X, Y\}$. Step 4 requires $O(|\mathbf{Z}|^2)$ tests. Step 5 requires $O(|\mathbf{Z}|)$ and Step 6 requires $O(|\mathbf{Z}|^2)$. Step 7 requires no tests, as it uses cached test results. Step 8 requires $O(|\mathbf{Z}|)$ tests. Thus, total tests performed is in $O(|\mathbf{Z}|^2)$. Empirical results with an oracle corroborate asymptotic analyses (Figure 5).

**Sample Complexity** The sample complexity of LDP will be dictated by the user-selected independence test. In lieu of a formal complexity analysis, we provide some statistical intuition for the efficiency of LDP. It is generally assumed that lower order CI relations (i.e., those with smaller conditioning sets) are inferred more reliably than higher order relations under finite samples [Spirtes et al., 2000]. For example, conditional mutual information (CMI) has been shown to require an exponential number of samples in the cardinality of the conditioning set [Kubkowski et al., 2021]. For a conditioning set of size 10 and a power of at least 0.5, CMI can require a sample size of approximately $n = 30\text{k}$ [Kubkowski et al., 2021]. Even if all variables in this conditioning set have just three discrete states, it is possible that only a subset of the $3^{10}$ possible states will be instantiated in sample sizes representative of real-world data [Spirtes et al., 2000]. Therefore, LDP was designed under the intuition that lower order CI tests provide more favorable sample complexity. The maximum conditioning set size for Line 12 is $O(|\mathbf{Z}_{1,2,3,5}|)$ and for the $\mathbf{Z}_5$ criterion is $O(|\mathbf{Z}_1|)$. All other conditioning sets are cardinality one or two. Limited

empirical comparisons by sample size (Figures 6, 7, A.1, A.2) and conditioning set size (Appendix G.1) are provided.

**LDP for VAS Selection Under Multiple Criteria** LDP flexibly facilitates VAS selection under multiple popular theoretical criteria (Appendix C). As LDP returns $\mathbf{Z}_1$, $\mathbf{Z}_4$, and $\mathbf{Z}_5$, LDP can be used as an automated preprocessing step for VAS selection under the *common cause criterion* (which retains only $\mathbf{Z}_1$), the *disjunctive cause criterion* (which retains $\{\mathbf{Z}_1, \mathbf{Z}_4, \mathbf{Z}_5\}$) [VanderWeele and Shpitser, 2011], and the *outcome criterion* (which retains $\{\mathbf{Z}_1, \mathbf{Z}_4\}$) [Brookhart et al., 2006], all of which yield VAS under the backdoor criterion and the *generalized adjustment criterion* [Perkovic et al., 2015]. However, we recommend caution if adjusting for $\mathbf{Z}_5$, as adjusting for instruments can amplify bias or introduce new bias [Pearl, 2012b]. We discuss notions of optimal and minimal adjustment sets in Appendix C.

### 4.1 IDENTIFIABILITY CONDITIONS

Here, we describe two separate sets of sufficient conditions for the identifiability of 1) a VAS for the true DAG and 2) correct partition labels, which are provided as intermediate results by LDP. Notably, we show that a VAS is identifiable under causal insufficiency if a specific CI criterion is met by at least one variable (Line 29 of Algorithm 1). All theoretical results are for the asymptotic regime and assume an independence oracle.

**Assumptions** We assume the causal Markov condition, faithfulness, and acyclicity. Variables are not assumed to be exclusively pretreatment and we do not place sparsity constraints on the true graph. We do not make assumptions about the distributional forms of variables nor the functional forms of their causal relations. While user-specified independence tests might impose their own parametric assumptions, nonparametric tests are recommended when the data generating process is unknown (e.g., Gretton et al. 2005, 2007, Zhang et al. 2011, Runge 2018).

**The Exposure-Outcome Pair** The only prior knowledge of $\mathcal{G}_{XY\mathbf{Z}}$ that is required by LDP concerns the exposure-outcome relationship. While the causal effect of $X$ on $Y$

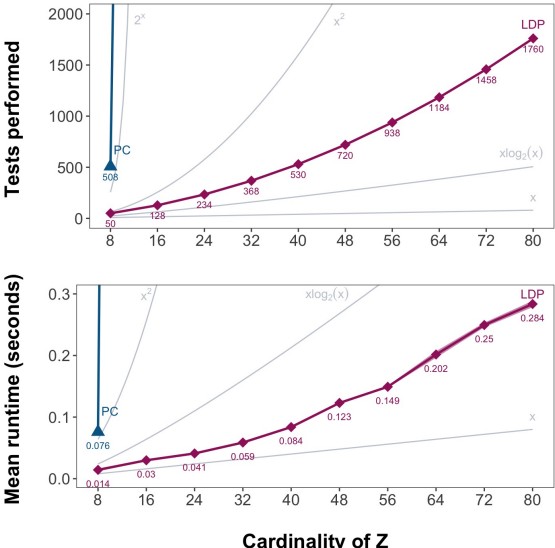

Figure 5: Total tests performed under an independence oracle (top) and mean runtime over 100 replicates (bottom) as the cardinality of $\mathbf{Z}$ increases, with 95% confidence intervals in shaded regions. Each DAG resembles Figure 3 with equal cardinality per partition ($[1, 10]$). Results are reported for LDP and PC. LDECC and MB-by-MB curves overlapped with PC, with PC outperforming. Exponential, quadratic, $x \log_2(x)$, and linear curves (in tests and milliseconds) serve as comparison. Table G.1 reports raw data.

can be of arbitrary strength or null, we assume that 1) $X$ and $Y$ are marginally dependent and 2) $Y$ cannot be a direct nor indirect cause of $X$ due to the acyclicity assumption. All proofs and experiments assume univariate $X$ and $Y$.

**Sufficient Conditions for VAS Identification**   A VAS 1) contains no descendants of $X$ and 2) blocks all backdoor paths for $\{X, Y\}$ (Definiton 2.4). Theorem 4.5 shows that the VAS returned by LDP meets both criteria given the following sufficient (but not necessary) graphical conditions.

C1  The existence of at least one observed member of $\mathbf{Z}_4$.
C2  The existence of at least one observed member of $\mathbf{Z}_5$, such that all $\mathbf{Z}_1$ are marginally independent of at least one observed $Z_5 \in \mathbf{Z}_5$.

C1 is testable at Line 7 of Algorithm 1 and C2 is testable in Steps 7 and 8. C1 guarantees that all backdoor paths will be blocked by the conditioning set in Step 5 of Algorithm 1 ($X \cup \mathbf{Z}' \backslash Z$), which is used to discover $\mathbf{Z}_5$. C2 guarantees that LDP identifies the $\mathbf{Z}_1$ needed to ensure a VAS, and enables the $\mathbf{Z}_5$ criterion to be tested (Definition 4.3). We note that C1 and C2 can be intuitively checked by reasoning whether *multiple causes of $X$ and $Y$* exist in the dataset. While the existence of multiple causes does not guarantee that these conditions will hold, the absence of multiple causes indicates that LDP might not be suitable. Verifying C1 does

not require C2 nor causal sufficiency (Theorem 4.6) and replaces the strong pretreatment assumption in the covariate selection literature. C1 is discussed further in Remark 4.7.

With respect to preventing descendants of $X$ from entering the adjustment set, we present Lemma 4.2.

**Lemma 4.2.** *LDP does not place descendants of $X$ in $\mathbf{Z}_1$ under Conditions C1 and C2.*

Additionally, LDP provides an internal test that indicates whether an adjustment set blocks all backdoor paths.

**Definition 4.3** ($\mathbf{Z}_5$ criterion). If there exists a $Z_5 \in \mathbf{Z}_5$ that is $d$-separable from $Y$ given $X$ and $\mathbf{Z}_1$ ($Z_5 \perp\!\!\!\perp Y | X \cup \mathbf{Z}_1$), we say that the $\mathbf{Z}_5$ criterion is passed.

**Lemma 4.4** (Passing the $\mathbf{Z}_5$ criterion is a valid indicator that $\mathbf{Z}_1$ blocks all backdoor paths). *If the $\mathbf{Z}_5$ criterion is passed, then the $\mathbf{Z}_1$ recovered by LDP is asymptotically guaranteed to block all backdoor paths for $X$ and $Y$.*

**Theorem 4.5** (LDP returns a VAS for $\{X, Y\}$ under the backdoor criterion). *Following from Lemmas 4.2 and 4.4, if the $\mathbf{Z}_5$ criterion is passed, then the $\mathbf{Z}_1$ returned by LDP is a VAS for $\{X, Y\}$.*

All proofs are provided in Appendix D.2. We numerically validate Theorem 4.5 in Section 6.

**Sufficient Conditions for Correct Partition Labels**   As we show in Theorem 4.6, Conditions C1 and C2 are *not* required to correctly label $\mathbf{Z}_4$, $\mathbf{Z}_7$, and $\mathbf{Z}_8$.

**Theorem 4.6.** *Partitions $\mathbf{Z}_4$, $\mathbf{Z}_7$, and $\mathbf{Z}_8$ are guaranteed to be correctly labeled by LDP in random structures without C1 and C2, even in the presence of latent confounding.*

*Intuition.*   Theorem 4.6 follows from the fact that tests for $\mathbf{Z}_4$, $\mathbf{Z}_7$, and $\mathbf{Z}_8$ rely only on knowledge of $\{X, Y\}$ and candidate $Z$. This is in contrast to Step 5, for example, which relies on access to additional variables in the true graph for correctness. Full proof is provided in Appendix D.3.

We note that LDP correctly labels partitions $\mathbf{Z}_1$, $\mathbf{Z}_2$, $\mathbf{Z}_3$, and $\mathbf{Z}_5$ under additional conditions C3 and C4. Given sufficient (but not necessary) conditions C1–C4, Theorem D.1 states that LDP correctly labels the causal partitions of $\mathbf{Z}$ as intermediate results en route to identifying a VAS (proof in Appendix D.1). Recall that C3 and C4 are not needed for identifying a VAS, nor partitions $\mathbf{Z}_4$, $\mathbf{Z}_7$, and $\mathbf{Z}_8$. Further, Section 6 provides empirical examples where C3 and C4 are violated with no impact on partition label accuracy for $\mathbf{Z}_1$, $\mathbf{Z}_2$, $\mathbf{Z}_3$, and $\mathbf{Z}_5$.

*Remark* 4.7 (Weakening the pretreatment assumption with Condition C1).   We argue that the common pretreatment requirement assumes away the problem of confounder ($\mathbf{Z}_1$) and instrument ($\mathbf{Z}_5$) identification via *a priori* exclusion of

$\{\mathbf{Z}_{2\in de(X)}, \mathbf{Z}_3, \mathbf{Z}_6, \mathbf{Z}_7\}$. We introduce Condition C1 based on the intuition that assuming the presence of at least one verifiable representative from a single partition ($\mathbf{Z}_4$) is more moderate than assuming the complete absence of multiple partitions, which may not be verifiable. We argue that C1 is a verifiable assumption, as we show that the correct identification of $\mathbf{Z}_4$ is robust to latent confounding in arbitrary DAGs (Theorem 4.6). Note that C1 is *not necessary* when $\mathbf{Z}$ contains no colliders (or, more strongly, when $\mathbf{Z}$ is pretreatment), nor when the $\mathbf{Z}_5$ criterion is passed (indicating that backdoor paths for $\{X, Y\}$ were closed despite failure to test for colliders; Lemma 4.4).

## 5 COMPARISON TO PRIOR METHODS

**Global Causal Discovery** While global methods can theoretically identify the partitions of $\mathbf{Z}$, LDP was explicitly designed to circumvent common drawbacks. The local approach of LDP avoids costly combinatorial optimization, guaranteeing worst-case polynomial test totals without sparsity constraints (at the expense of graphical assumptions C1 and C2). Further, the asymptotic guarantees of nonparametric global discovery can fail even on simple structures under small to moderately large samples, which are common in practice (Figures A.1–A.3). LDP addresses sample complexity by favoring lower order CI tests relative to global constraint-based methods [Spirtes et al., 2000].

**Local Discovery Around Target Variables** The challenges of global discovery can be mitigated by local methods that infer relevant substructures around a target (or targets) of interest. Most local methods for causal ancestor discovery, confounder discovery, or related tasks impose strong graphical assumptions that require prior knowledge. The most common of these assumptions requires that input variables are non-descendants of the target (e.g., the *pretreatment assumption* in the exposure-outcome context) [De Luna et al., 2011, Entner et al., 2013, Häggström et al., 2015, Shortreed and Ertefaie, 2017, Tian et al., 2018, Gultchin et al., 2020, Soleymani et al., 2022, Shah et al., 2022, Cai et al., 2023]. We argue that excluding the existence of colliders, mediators, and other descendants of the exposure overly simplifies the problem of identifying instruments, confounders, and other variables that are useful for downstream inference.

**Automated Covariate Selection for Pretreatment $\mathbf{Z}$** Entner et al. [2013], Gultchin et al. [2020], Shah et al. [2022], Cheng et al. [2023b], and Cheng et al. [2023a] assume the existence of *anchor* or *auxiliary variables*, which can resemble $\mathbf{Z}_1$ or $\mathbf{Z}_5$. Thus, this assumption plays a similar role to Condition C2 and Lemma 4.4. With the exception of Cheng et al. [2023a], these methods require pretreatment $\mathbf{Z}$, a strong assumption that we significantly weaken by introducing Condition C1. While Cheng et al. [2023b] require that auxiliary variables were identified prior to confounder

discovery, LDP discovers the variables needed to satisfy Conditions C1 and C2 end-to-end. The continuous optimization approach taken by Gultchin et al. [2020] requires parametric assumptions, while LDP does not.

We should emphasize that these methods are designed for VAS discovery alone, sometimes combined with causal effect estimation end-to-end. LDP, on the other hand, is a *local discovery procedure* for partitioning $\mathbf{Z}$ while guaranteeing a VAS. Thus, unlike prior methods, LDP can be used to satisfy multiple covariate selection criteria (Section 4) or to assist with tasks beyond valid adjustment (e.g., discovering instruments and their proxies, causes of outcome, etc.).

**Automated Covariate Selection for Arbitrary $\mathbf{Z}$** Like LDP, concurrent work by Cheng et al. [2023a] avoids the pretreatment assumption. However, it is not a local method as it calls an existing variant of FCI [Colombo et al., 2012], a global discovery algorithm. Lemma 4.4 bears similarity to Theorem 1 in Cheng et al. [2023a], in which they prove that an analogous CI relation indicates the existence of VAS in partial ancestral graphs with hidden variables.

## 6 EXPERIMENTAL RESULTS

We numerically validate LDP on custom synthetic DAGs and the MILDEW benchmark from the `bnlearn` Bayesian Network Repository (Figure E.3) [Scutari, 2010].[6] For all simulated DAGs, structural equations are reported in Tables G.2 and G.3 and graphs are visualized in Appendix E. Experimental data generation is described in Appendix F. We experimentally validate LDP for VAS discovery in causally sufficient $\mathbf{Z}$ and in the presence of latent confounding. Additionally, we probe 1) partition label correctness and 2) the quality of adjustment sets ($\mathbf{A}_{XY}$) returned by each method with respect to average treatment effect (ATE) estimation.

**Baseline Discovery Methods** All baselines are constraint-based and do not assume pretreatment. PC and FCI are global discovery algorithms with asymptotic theoretical guarantees and worst-case exponential time complexity with respect to node count [Spirtes et al., 2000]. Two local methods were also selected for comparison. MB-by-MB [Wang et al., 2014] and Local Discovery Using Eager Collider Checks (LDECC) [Gupta et al., 2023] take distinct approaches to inferring the local structure around a target node. While MB-by-MB is exponential-time, LDECC is provably polynomial-time for certain categories of graphs and exponential for others. Further description of these algorithms and how they were evaluated is in Appendix F. To illustrate the strengths and weaknesses of all baselines, VAS were evaluated under the *common cause criterion* (CCC) [Guo et al., 2022] and *disjunctive cause criterion* (DCC) [VanderWeele and Shpitser, 2011] (Appendix C).

---

[6]https://www.bnlearn.com/bnrepository/

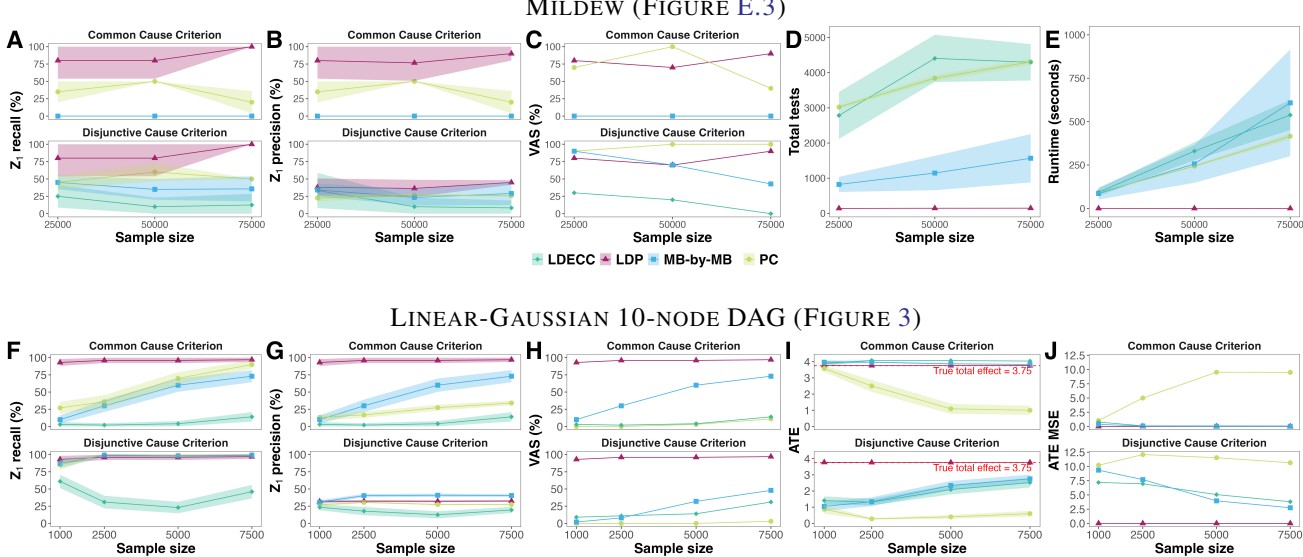

Figure 6: Baselines on MILDEW ($|\mathbf{Z}| = 31$) and a linear-Gaussian DAG ($|\mathbf{Z}| = 8$) (Tables G.8, G.9). Independence was determined with chi-square tests for MILDEW ($\alpha = 0.001$) and Fisher-z tests for the linear-Gaussian DAG ($\alpha = 0.01$). Results were averaged over 10 and 100 replicates per sample size for MILDEW and the linear-Gaussian DAG, respectively (95% confidence intervals in shaded regions). Precision and recall for $\mathbf{Z}_1$ identification were computed per adjustment set.

**Partition Accuracy** We measure partition accuracy as the percentage of partition labels that are consistent with ground truth. Results on the 10-node DAG with one variable per partition (Figure 3) indicate that LDP correctly partitions $\mathbf{Z}$ under continuous, discrete, linear, and nonlinear data generating processes (Figure G.1, Tables G.4, G.5). Figure G.1 supports the claim that LDP is agnostic to the strength of the direct effect of $X$ on $Y$, as results are unharmed when $X$ is not adjacent to $Y$. Though LDP is not guaranteed to correctly partition when inter-partition active paths exist, we demonstrate that LDP is robust to certain violations of this condition (Tables G.6, G.7, G.10): LDP provides high partition accuracies on the MILDEW benchmark ($\geq 90\%$ accuracy; Figure G.2) and synthetic DAGs when $\mathbf{Z}_2$ shares active paths with $\mathbf{Z}_4$, $\mathbf{Z}_5$, and $\mathbf{Z}_6$ (Figures E.1, E.2, E.5).

## 6.1 VAS UNDER CAUSAL SUFFICIENCY

We compare adjustment set quality across baselines for two graphs with small $\mathbf{Z}_1$ (Figure 6): the MILDEW benchmark ($|\mathbf{Z}| = 31; |\mathbf{Z}_1| = 2$) and a linear-Gaussian DAG ($|\mathbf{Z}| = 8; |\mathbf{Z}_1| = 1$). Quality was measured in terms of $\mathbf{Z}_1$ precision and recall in $\mathbf{A}_{XY}$, percent of $\mathbf{A}_{XY}$ that were valid, total tests performed, runtime, and ATE MSE (when ground truth was available). Results indicate that LDP provides superior sample, statistical, and time efficiency relative to baselines.

LDP outperformed on $\mathbf{Z}_1$ recall (Figure 6.A, 6.F top), test count (6.D), and runtime (6.E). Notably, LDP ran $1400\times$ to $2500\times$ faster than PC across sample sizes for MILDEW, with comparable gains relative to local baselines (Table G.8).

High $\mathbf{Z}_1$ recall for LDP is reflective of its ability to detect $\mathbf{Z}_1$ that are not adjacent to either $X$ nor $Y$, unlike local baselines. As expected, LDP displayed superior $\mathbf{Z}_1$ precision under the CCC but was comparable to other methods when $\mathbf{Z}_4$ and $\mathbf{Z}_5$ were intentionally retained under the DCC (6.B, 6.G). Only LDP consistently returned a VAS for the linear-Gaussian DAG under the CCC and DCC (6.H). Furthermore, $\mathbf{A}_{XY}$ from LDP provided less biased and more precise ATE estimates (6.I, 6.J). Highly biased ATE estimates using $\mathbf{A}_{XY}$ from PC is linked to a propensity to include extraneous variables (Figure G.3). Low ATE variance for LDP implies favorable statistical efficiency relative to baselines. Further, LDP achieves consistently high VAS quality at smaller sample sizes than baselines, implying greater sample efficiency.

Additionally, we illustrate a known failure mode of LDP partition labeling that still results in VAS (Figure E.4; $|\mathbf{Z}| = 14; |\mathbf{Z}_1| = 7$). In a complex backdoor path, a $Z_1$ adjacent to $Y$ is marginally dependent on a $Z_4$ and will be mislabeled as $\mathbf{Z}_{\mathrm{POST}}$. Further, a $Z_2$ that is 1) a non-descendant of $X$ and 2) conditionally independent of $\{X, Y\}$ given $\mathbf{Z}_1$ is guaranteed to be placed in $\mathbf{Z}_1$. Despite these mislabelings, LDP returned a VAS for 99% of 100 replicates (sample size $n = 5\mathrm{k}$). Figure E.4 describes further details.

## 6.2 VAS DISCOVERY WITH LATENT VARIABLES

With the $\mathbf{Z}_5$ criterion (Line 29 of Algorithm 1; Definition 4.3), LDP helps the user to manage uncertainty about the quality of the returned adjustment set. To numerically val-

| Latent | VAS Exists | $\mathbf{Z}_5$ Crit | % Valid |
|---|---|---|---|
| $B_1 \in \mathbf{Z}_1$ | ✓ | ✓ | 100 |
| $B_2 \in \mathbf{Z}_1$ | ✓ | ✓ | 99 |
| $Z_{4a} \in \mathbf{Z}_4$ | ✓ | ✓ | 99 |
| $M_2 \in \mathbf{Z}_4$ | ✓ | ✓ | 100 |
| $Z_{5a} \in \mathbf{Z}_5$ | ✓ | ✓ | 99 |
| $M_1 \in \mathbf{Z}_5$ | ✓ | ✓ | 100 |
| $Z_1 \in \mathbf{Z}_1$ | ✗ | ✗ | 0 |
| $B_3 \in \mathbf{Z}_1$ | ✗ | ✗ | 0 |

Table 4: Numerical validation of Theorem 4.5 on an 18-node ground truth DAG with latent variables (Figures 7, E.5). For 100 replicates where each variable was left unobserved (Latent), we report whether a VAS for the ground truth DAG exists in $\mathbf{Z}$ (VAS Exists), whether the $\mathbf{Z}_5$ criterion passed ($\mathbf{Z}_5$ Crit), and the percent of predicted adjustment sets that were valid with respect to the ground truth DAG (% Valid). Additional information is provided in Table G.10.

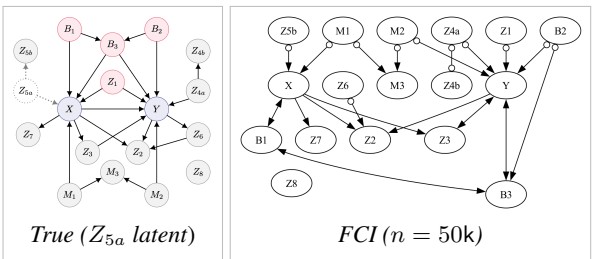

*True ($Z_{5a}$ latent)*  *FCI ($n = 50$k)*

Figure 7: FCI failed to identify a VAS on DAGs with linear causal functions and Bernoulli noise (true $\mathbf{Z}_1$ in red; 5/5 replicates consistent with the predicted DAG at right). Additional results for PC and FCI are reported in Figure A.3.

idate the asymptotic guarantees of this criterion and Theorem 4.5, we ran LDP with an independence oracle on causally insufficient structures. The ground truth DAG contained a butterfly structure, M-structure, and inter-partition active paths (Figures 7, E.5; $|\mathbf{Z}| = 18$; $|\mathbf{Z}_1| = 4$). All combinations of common causes (up to size three) were iteratively dropped from the observed data. Hidden confounders were in $\mathbf{Z}_1$, $\mathbf{Z}_4$, and $\mathbf{Z}_5$ and induced latent confounding between $\{X, Y\}$, $\{X, \mathbf{Z}_1\}$, $\{Y, \mathbf{Z}_1\}$, $\{X, \mathbf{Z}_2\}$, $\{Y, \mathbf{Z}_2\}$, $\{Y, \mathbf{Z}_4\}$, and $\{X, \mathbf{Z}_5\}$. Partition $\mathbf{Z}_2$ shared active paths with $\mathbf{Z}_4$, $\mathbf{Z}_5$, and $\mathbf{Z}_6$.

For all eight structures with one latent variable, we observed 100% concordance between LDP passing the $\mathbf{Z}_5$ criterion, whether a VAS for the true DAG existed in $\mathbf{Z}$, and whether LDP returned a VAS. For the 84 structures with two to three latent variables, we saw 100% concordance between whether the $\mathbf{Z}_5$ criterion was passed and whether LDP returned a VAS. In all instances (6%) where a VAS existed in $\mathbf{Z}$ and LDP failed to return a VAS, both parents of the M-collider were latent. As expected, such a M-collider was treated as $\mathbf{Z}_1$. In all such cases, LDP raised a warning that

the $\mathbf{Z}_5$ criterion was failed and a VAS was not identified. These results suggest that the $\mathbf{Z}_5$ criterion is a valid indicator for whether the $\mathbf{Z}_1$ returned by LDP is a VAS and, further, whether it induces *M-bias* [Ding and Miratrix, 2014].

To probe finite sample performance, we ran LDP on linear categorical instantiations of Figure E.5 ($n = 50$k; chi-square tests; $\alpha = 0.001$). We tested 100 replicates per latently confounded structure. Among 600 instances for which a VAS of the ground truth DAG existed in $\mathbf{Z}$, LDP returned a VAS for 99.5% (95% CI [99.1, 99.9]; Table 4). If at least one parent in the M-structure was observed in $\mathbf{Z}$, the collider was not placed in $\mathbf{Z}_1$ and LDP returned a VAS without M-bias (200/200 replicates).

In contrast, PC and FCI demonstrate finite sample failure modes on the same causal structure (Figures 7, A.3). Though PC and FCI return a VAS when provided with an oracle, both algorithms fail to provide VAS for discrete and continuous data samples (Figure A.3). In particular, these methods display a high false negative rate on true confounder $Z_1$.

# 7 LIMITATIONS AND FUTURE WORK

The performance of LDP will be constrained by the accuracy, runtime, and sample complexity of the chosen independence test. While LDP does not make innate parametric assumptions, the user should be cautious if opting for parametric independence tests. We provide asymptotic theoretical guarantees, which future work could extend to probabilistic guarantees under finite samples.

As causal discovery is notoriously impractical in many settings, this work attempts to highlight the benefits of approaches that are tailored for specific use cases. Under a tailored approach, prior knowledge of the problem space can improve performance relative to generalized global discovery (e.g., in time and sample efficiency).

In this work, we propose a performant local discovery algorithm for VAS discovery, at the expense of Conditions C1 and C2. These graphical conditions restrict the space over which LDP provides informative results. We hope to see performant local discovery solutions to the covariate selection problem that do not assume pretreatment yet do not rely on the presence of $\mathbf{Z}_4$ and $\mathbf{Z}_5$. In particular, the efficient differentiation of confounders, mediators, and colliders in random graphs is a challenging problem that we hope future research will address.

## ACKNOWLEDGEMENTS

The authors would like to acknowledge support from the NSF Graduate Research Fellowship (author J. Maasch); NSF 1750326, 2212175; and NIH R01AG080991, R01AG076234.

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

# APPENDIX

## A MOTIVATING EXAMPLES: COMPARISON TO GLOBAL CAUSAL DISCOVERY

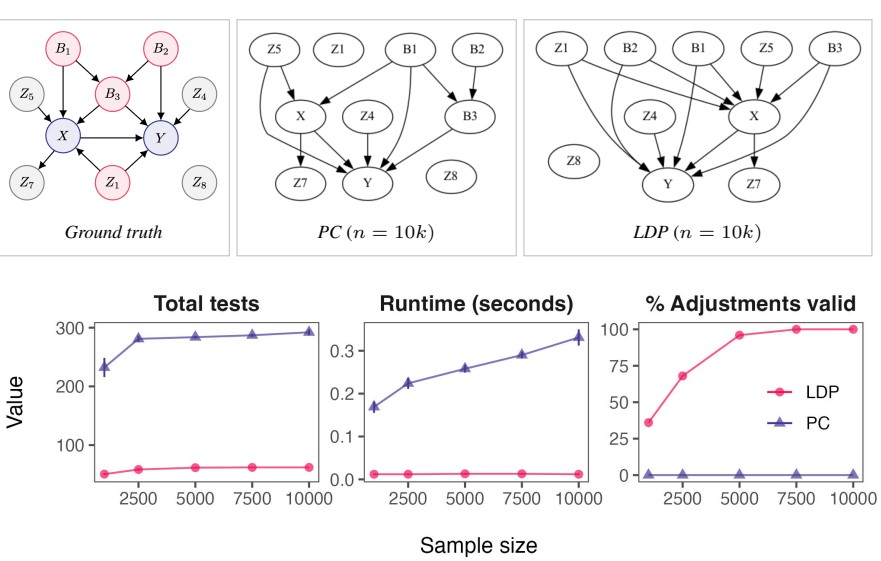

Figure A.1: Mean independence tests performed, mean runtime (seconds), and percent of adjustment sets that were valid for the experiments described in Figure 1. Values were averaged over 25 replicates per sample size for a linear-Gaussian DAG (Fisher-z tests; $\alpha = 0.005$). Error bars represent 95% confidence intervals. Experiments used the PC implementation by Kalisch and Buhlmann [2007]. Note that LDP does not infer the relations between members of $\mathbf{Z}$, hence only the paths to $X$ and $Y$ are visualized (abstracted as length-1).

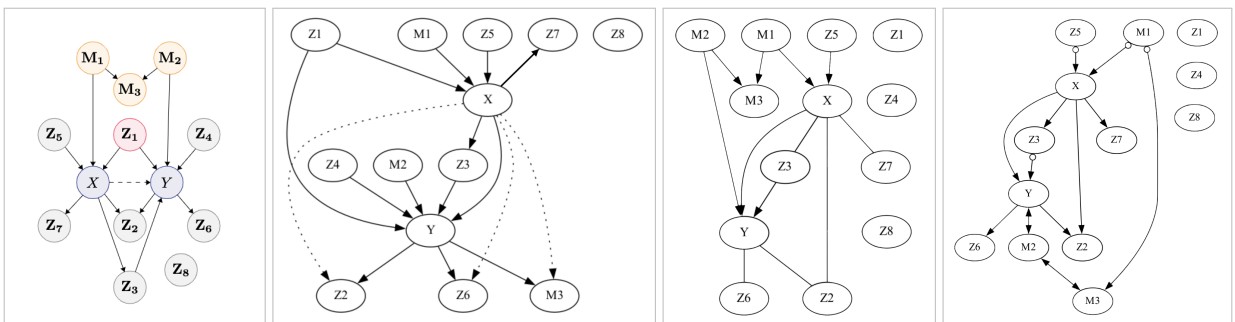

Figure A.2: A ground truth DAG with M-structure versus the DAGs inferred by LDP, PC [Spirtes et al., 2000], and FCI [Spirtes et al., 2000] (left to right). Causal mechanisms were linear and noise was Bernoulli ($n = 5k$; chi-square independence tests). Only LDP partitioned the variables correctly and returned a valid adjustment set (VAS). LDP returned a VAS at $n = 1k$, FCI at $n = 10k$, and PC at $n > 10k$. Note that LDP does not infer the relations between members of $\mathbf{Z}$, hence only the paths to $X$ and $Y$ are visualized (abstracted as length-1). Dotted edges indicate that LDP could not distinguish $\mathbf{Z}_2$ from $\mathbf{Z}_6$, which is expected behavior on this structure. Variables $M_1 \in \mathbf{Z}_5$, $M_2 \in \mathbf{Z}_4$, and $M_3 \in \mathbf{Z}_2$. Experiments use the PC implementation from [Kalisch and Buhlmann, 2007] and the FCI implementation from `causal-learn`.

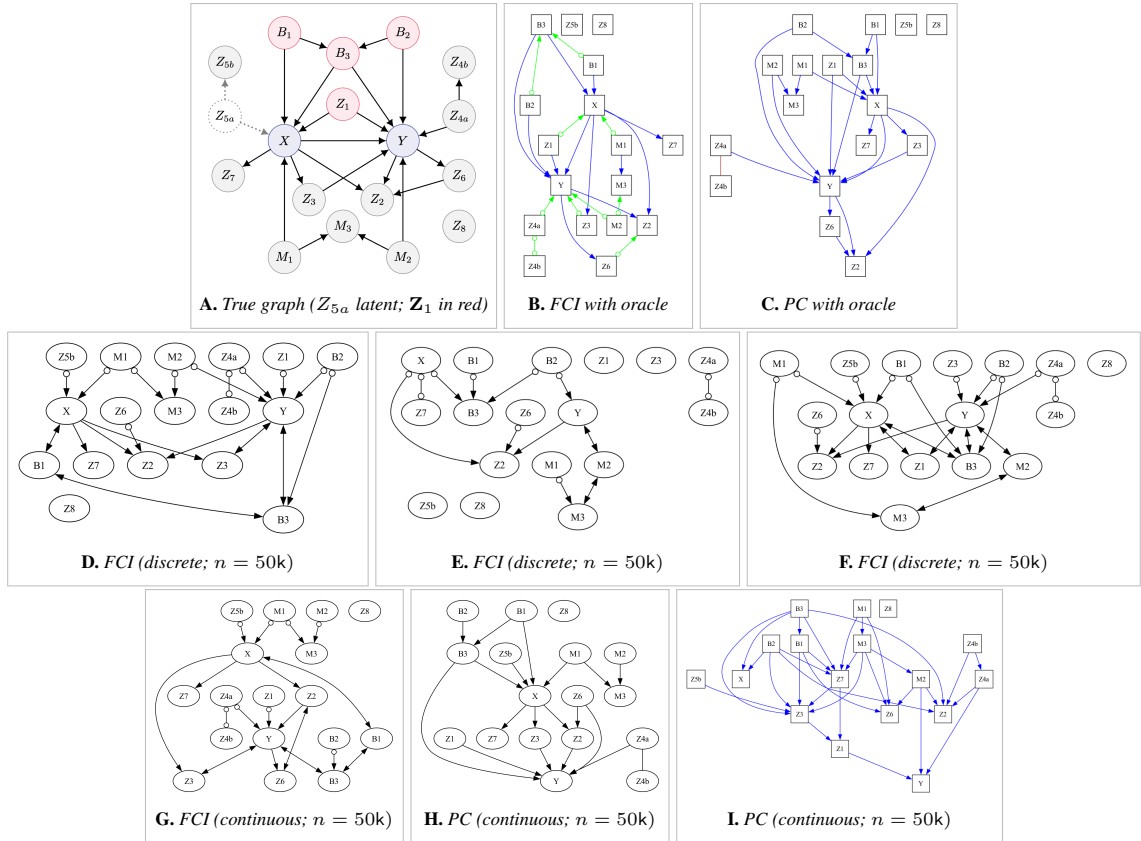

Figure A.3: Failure modes of PC and FCI for causal partitioning under finite samples. Though PC and FCI return a valid adjustment set (VAS) when provided with an oracle (**B**, **C**), both algorithms fail to provide VAS for discrete and continuous data samples. Though the correlation coefficient for $X$ and $Z_1$ was moderate for all examples ($\in [0.25, 0.36]$) and both parametric and nonparametric (mutual information) tests demonstrated marginal dependence, neither algorithm could reliably detect the edge between $X$ and $Z_1$ (**D**, **E**). Even when FCI inferred edges between $Z_1$, $X$, and $Y$ (**F**), the bidirected edges $Y \leftrightarrow Z_1 \leftrightarrow X$ imply that $Z_1$ is an M-collider, not a confounder. Unreliable results might be attributable to cascading errors due to the ordering of tests [Colombo and Maathuis, 2014]. Discrete data (**D–F**) feature linear causal mechanisms and Bernoulli noise (chi-square tests; $\alpha = 0.001$). Continuous data (**G–I**) feature linear causal mechanisms and Gaussian noise (Fisher-z tests; $\alpha = 0.001$). Varying $\alpha$ did not improve causal partitioning. LDP successfully identified a VAS for both discrete and continuous instantiations. Graphs produced with `dodiscover` (**B**, **C**, **I**) and `causal-learn` (**D–H**).

# B  CAUSAL PARTITIONS: EXTENDED DEFINITIONS

## B.1  PARTITION $Z_5$: INSTRUMENTAL VARIABLES AND THEIR PROXIES

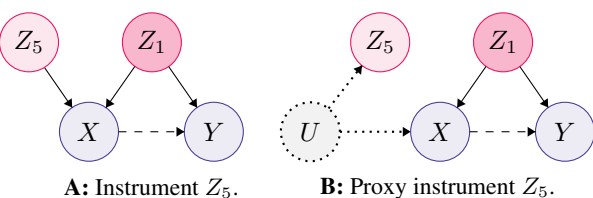

**A:** Instrument $Z_5$.    **B:** Proxy instrument $Z_5$.

Figure B.1: $\mathbf{Z}_5$ encompasses (**A**) instruments that are causal for $X$ and (**B**) proxy instruments that are descended from confounders $U \in \mathbf{Z}_5$ (i.e., common causes of proxy instruments and $X$) [Hernán and Robins, 2006].

Instrumental variable methods have been used heavily in econometrics [Imbens, 2014] and epidemiology [Hernán and Robins, 2006, Labrecque and Swanson, 2018] for causal effect estimation in the presence of latent confounding. The

present work explores an additional way to relate instrumental variables to the problem of confounding, where the marginal independence between some instrument-confounder pairs is exploited to detect confounders in unknown causal structures. We define an instrument as any variable that meets the criteria enumerated in Definition B.1. We then claim Proposition B.2 about the relations among $\mathbf{Z}_1$ and $\mathbf{Z}_5$, as a theoretical basis for sufficient condition C2. Proof of Proposition B.2 follows from Propositions B.5 and B.7.

**Definition B.1** (Instrumental variable, Lousdal [2018]). Any instrument meets the following criteria:

1. *Relevance assumption*: The instrument is causal for exposure $X$.
2. *Exclusion restriction*: The effect of the instrument on outcome $Y$ is fully mediated by $X$.
3. *Exchangeability assumption*: The instrument and $Y$ do not share a common cause.

**Proposition B.2.** *Any instrument (or proxy) $Z_5 \in \mathbf{Z}_5$ will meet the following criteria with respect to at least one confounder (or proxy) $Z_1 \in \mathbf{Z}_1$ on every backdoor path in $\mathcal{G}_{XY\mathbf{Z}}$.*

1. *$Z_5$ and $Z_1$ are marginally independent.*
2. *$Z_5$ and $Z_1$ are conditionally dependent given $X$.*

## B.2 PARTITION $\mathbf{Z}_4$

To our knowledge, partition $\mathbf{Z}_4$ has been significantly less characterized and less utilized in the causal inference literature than confounders ($\mathbf{Z}_1$), colliders ($\mathbf{Z}_2$), mediators ($\mathbf{Z}_3$), and instrumental variables ($\mathbf{Z}_5$). Limited reference has been made to members of this partition under the term *pure prognostic variables* [Hahn and Herren, 2022]. We elaborate on our definition of $\mathbf{Z}_4$ below.

**Definition B.3** (Partition $\mathbf{Z}_4$). Partition $\mathbf{Z}_4$ encompasses all non-descendants of $Y$ that are marginally dependent on $Y$ but marginally independent of $X$ (Table 1). Given this definition, we observe that any $Z_4 \in \mathbf{Z}_4$ participates in a $v$-structure $X \cdots \to Y \leftarrow \cdots Z_4$. This implies the following:

1. $X$ cannot share active paths with any $Z_4$. Thus, $X$ can share no common causes with any $Z_4$.
2. $\mathbf{Z}_4$ is conditionally dependent on $X$ given $Y$. This implicitly requires that the $X$ and $Y$ under consideration are marginally dependent (an assumption made in Section 4.1), though they may not be directly adjacent in $\mathcal{G}_{XY\mathbf{Z}}$.

## B.3 ADDITIONAL PROPOSITIONS ON $\mathbf{Z}_4$ AND $\mathbf{Z}_5$

Here, we introduce several propositions that describe the properties of $\mathbf{Z}_4$ and $\mathbf{Z}_5$ in relation to each other and to $\mathbf{Z}_1$. Let $\mathcal{P}$ be a backdoor path in $\mathcal{G}_{XY\mathbf{Z}}$.

**Proposition B.4.** *If a $Z_4 \in \mathbf{Z}_4$ shares an active path with any $Z_1 \in \mathbf{Z}_1$ on $\mathcal{P}$ such that $Z_4 \not\perp\!\!\!\perp Z_1$, that $Z_4$ must form a $v$-structure $Z_4 \cdots \to Z_1 \leftarrow \cdots Z_1'$, where $Z_1'$ lies between $Z_1$ and $X$ on $\mathcal{P}$. If not, $Z_4$ would share an active path with $X$, which violates the definition of $\mathbf{Z}_4$ (Definition B.3). In Figure B.2 (right-hand DAG), examples include $Z_4 \to Z_1^3 \leftarrow Z_1^2$ and $Z_4 \to Z_1^3 \leftarrow Z_1^5$. Together with Definition B.3, this proposition implies that no $Z_4$ will ever be marginally dependent on a $Z_1$ that is directly adjacent to $X$.*

**Proposition B.5.** *If a $Z_5 \in \mathbf{Z}_5$ shares an active path with any $Z_1 \in \mathbf{Z}_1$ on $\mathcal{P}$ such that $Z_5 \not\perp\!\!\!\perp Z_1$, that $Z_5$ must form a $v$-structure $Z_5 \cdots \to Z_1 \leftarrow \cdots Z_1'$, where $Z_1'$ lies between $Z_1$ and $Y$ on $\mathcal{P}$. If not, $Z_5$ would share an active path with $Y$, which violates the definition of $\mathbf{Z}_5$ (Definition B.1). In Figure B.2 (right-hand DAG), examples include $Z_5 \to Z_1^1 \leftarrow Z_1^2$ and $Z_5 \to Z_1^1 \leftarrow Z_1^4$. Together with Definition B.1, this proposition implies that no $Z_5$ will ever be marginally dependent on a $Z_1$ that is directly adjacent to $Y$.*

**Proposition B.6** (A single $Z_1 \in \mathbf{Z}_1$ cannot be a collider for a $Z_4 \in \mathbf{Z}_4$ and a $Z_5 \in \mathbf{Z}_5$). *If a single $Z_1$ was a collider for $Z_4$ and $Z_5$, then $Z_4$ would share an active path with $X$ and $Z_5$ would share an active path with $Y$, violating the definitions of these partitions.*

Next, we introduce the concepts of *root-$Z_1$* and *collider-$Z_1$*. We observe that every backdoor path features a $Z_1$ that acts as a *root* node for that path: i.e., it is a common cause for $\{X, Y\}$ and all $Z_1$ that are its descendants on the paths to $X$ and $Y$. In Figure B.2, $\{Z_1^1, Z_1^3, Z_1^6\}$ are roots for backdoor paths in the left-hand DAG while $\{Z_1^2, Z_1^4, Z_1^5\}$ are roots for backdoor paths in the right-hand DAG. When multiple backdoor paths in $\mathcal{G}_{XY\mathbf{Z}}$ overlap (i.e., share subpaths), some $Z_1$ can behave as

*colliders* for two parents in $\mathbf{Z}_1$. In Figure B.2, $\{Z_1^2, Z_1^4\}$ are *collider-$Z_1$* on overlapping backdoor paths in the left-hand DAG while $\{Z_1^1, Z_1^2, Z_1^3\}$ are *collider-$Z_1$* for backdoor paths in the right-hand DAG. Note that node $Z_1^2$ in the right-hand DAG simultaneously behaves as a *root-$Z_1$* and a *collider-$Z_1$* for different backdoor paths. Thus, $Z_1^2$ is not a *true root* in the classical graph theory sense of having no parents.

**Proposition B.7** (The *root-$Z_1$* of a backdoor path will never be marginally dependent on a $Z_4$ nor a $Z_5$)**.** *As all root-$Z_1$ are causal for both $X$ and $Y$, marginal dependence on either a $Z_4$ or a $Z_5$ would violate Propositions B.4, B.5, and B.6.*

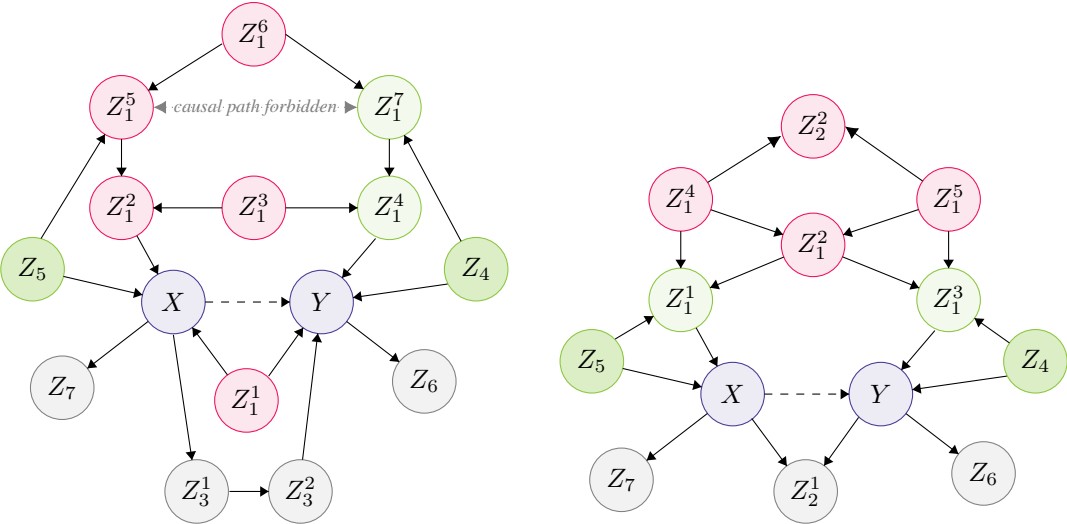

Figure B.2: Two DAGs that exemplify the behavior of LDP for valid adjustment set detection in the presence of inter-partition active paths. All red nodes will be placed in $\mathbf{Z}_1$ by LDP. All confounders for $\{X, Y\}$ that are colored green will be mislabeled due to their marginal dependence on $Z_4$ or $Z_5$.

***Left:*** Variables $Z_1^1$, $Z_1^3$ and $Z_1^6$ will be placed in $\mathbf{Z}_1$. Despite their marginal dependence on the only $Z_5$ in this structure, $Z_1^2$ and $Z_1^5$ will never be placed in $\mathbf{Z}_{\text{POST}}$ due to the presence of $Z_1^1$, as $Z_1^2 \perp\!\!\!\perp Z_1^1$ and $Z_1^5 \perp\!\!\!\perp Z_1^1$. Together, the confounders highlighted in red ($\{Z_1^1, Z_1^2, Z_1^3, Z_1^5, Z_1^6\}$) constitute a valid adjustment set that blocks all backdoor paths and contains no descendents of $X$. No causal path of either directionality is permissible between $Z_1^5$ and $Z_1^7$, as it would induce an active path from $Z_4$ to $X$ or from $Z_5$ to $Y$ not mediated by $X$. If this path were to contain a confounder analogous to $Z_1^3$, this would be permissible and this node would be placed in $\mathbf{Z}_1$ by LDP.

***Right:*** This DAG contains a modified butterfly structure, which will be partially retained in $\mathbf{Z}_1$ ($\{Z_1^2, Z_1^4, Z_1^5\}$) while still blocking all backdoor paths. As there is only one $Z_5$ in this structure and no backdoor path whose members are marginally independent of $Z_1^1$, this confounder will be mislabeled as $\mathbf{Z}_{\text{POST}}$ at Step 6. This DAG also illustrates a case where a member of $\mathbf{Z}_2$ ($Z_2^2$) is placed in $\mathbf{Z}_1$. Inclusion of $Z_2^2$ does not violate the validity of the adjustment set returned by LDP, as this node is not a descendent of $X$ and additionally adjusting for $\{Z_1^2, Z_1^4, Z_1^5\}$ prevents collider bias.

## B.4   PROXY VARIABLES

Multiple causal partitions defined in this work include notions of *proxy variables*. These proxies are conceptually related to previously described proxy variables in the causal literature, though they may depart in some ways. Firstly, the path types enumerated in Table 2 allow for proxies of confounders to be classified as $\mathbf{Z}_1$. A descendant proxy can act as a noisy stand-in for its respective confounder [Pearl, 2012a], and adjusting for this proxy when the confounder is unobserved can theoretically reduce confounding bias (though this is not guaranteed for all cases) [VanderWeele, 2019]. Likewise, proxy instruments are a notable variable type in the instrumental variable literature that falls under our definition of $\mathbf{Z}_5$ (Figure B.1). We generalize the notion of a proxy here to refer to any member of $\mathbf{Z}_1$ that does not lie on a backdoor path (and thus cannot fully block it), as well as the analogue for $\mathbf{Z}_2$ and $\mathbf{Z}_3$. For the purposes of this work, the proxy and the variable that it proxies will generally both be observed, though the literature explores cases where the proxied variable is unobserved [Wang and Blei, 2021].

**Definition B.8** (Proxy variables in $\mathbf{Z}_1$, $\mathbf{Z}_2$, and $\mathbf{Z}_3$)**.** A proxy variable for $\mathbf{Z}_1$, $\mathbf{Z}_2$, or $\mathbf{Z}_3$ is a member of these partitions that is an ancestor or descendant of another member of its respective partition, such that the proxy is not strictly a confounder,

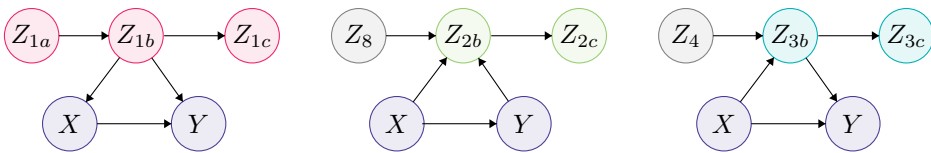

Figure B.3: Example proxy variables for $\mathbf{Z}_1$, $\mathbf{Z}_2$, and $\mathbf{Z}_3$. All $Z_{*a}$ and $Z_{*c}$ are proxies for the corresponding $Z_{*b}$.

mediator, or collider, but still satisfies the allowable path types for its respective partition (as defined in Table 3). This includes members of $\mathbf{Z}_3$ that are not directly on mediator chains but are descended from $\mathbf{Z}_3$ that lie on mediator chains, members of $\mathbf{Z}_1$ that are not on backdoor paths but are ancestral to $\mathbf{Z}_1$ on backdoor paths, etc. (Figure B.3).

## C   COVARIATE SELECTION CRITERIA

**Popular Criteria For Valid Adjustment**   The notion of valid adjustment is contingent on the causal quantity of interest. In this work, we consider valid adjustment with respect to total effect estimation. Pearl's *backdoor path criterion* dictates that a valid adjustment set contains no descendants of the exposure and blocks all backdoor paths (Definition 2.4) [Pearl, 1995]. Additional covariate selection criteria have been proposed, which are consistent with the backdoor criterion but provide additional guidance. The *common cause criterion* advocates controlling only for confounders ($\in \mathbf{Z}_1$), and is popular in practice [Guo et al., 2022]. The *pretreatment criterion* controls for all measured baseline variables, an approach previously defended by Donald Rubin [Rubin, 2008, Guo et al., 2022]. This approach is at risk of overadjustment [Schisterman et al., 2009, Lu et al., 2021] as it could allow instruments ($\mathbf{Z}_5$) and M-structure colliders ($M_3 \in \mathbf{Z}_2$; Figure E.1) to be included in the adjustment set (see Pearl 2012b, Ding and Miratrix 2014 for discussions of when this may be problematic). The *disjunctive cause criterion* is an intermediate approach between the common cause and pretreatment criteria [VanderWeele and Shpitser, 2011]. This criterion retains covariates that are causal for exposure, outcome, or both (i.e., $\mathbf{Z}_1$, $\mathbf{Z}_4$, and $\mathbf{Z}_5$). Adjusting only for $\mathbf{Z}_1$ and $\mathbf{Z}_4$ has also been advocated for propensity score models [Brookhart et al., 2006], as 1) unnecessarily adjusting for $\mathbf{Z}_5$ raises risks of variance inflation and bias amplification while 2) adjusting for $\mathbf{Z}_4$ can improve causal estimate precision without impacting bias. We refer to this approach as the *outcome criterion*. The *generalized adjustment criterion* [Perkovic et al., 2015] extends the *generalized backdoor criterion* [Maathuis and Colombo, 2015] to provide a unified criterion for necessary and sufficient adjustment sets that applies to DAGs, maximum ancestral graphs (MAGs), completed partially directed acyclic graphs (CPDAGs), and partial ancestral graphs (PAGs).

**Optimality and Minimality**   LDP returns the entire discovered $\mathbf{Z}_1$ and does not explicitly infer optimal nor minimal valid adjustment sets. LDP solves a *local causal discovery problem* and not an inference problem, and the notion of optimality is only well-defined for a particular estimator of the target parameter. However, LDP can be applied as an efficient preprocessing step for algorithms that return optimal and minimal adjustment sets but require prior graphical knowledge. Commonly, optimality is defined in terms of minimizing the asymptotic variance of causal estimates [Runge, 2021]. It has been shown that constraining the outcome by adjusting for $\mathbf{Z}_4$ in a propensity score model can decrease average treatment effect variance [Brookhart et al., 2006]. Since LDP discovers $\mathbf{Z}_4$ by design, it can be used for this objective.

# D PROOFS

In the following proofs, we assume that all assumptions and sufficient conditions defined in Section 4.1 are met unless it is explicitly stated that they can be weakened or dropped.

## D.1 PARTITION CORRECTNESS OF ALGORITHM 1

Per Theorem 4.6, LDP correctly labels partitions $\mathbf{Z}_4$, $\mathbf{Z}_7$, and $\mathbf{Z}_8$ without conditions C1 and C2, even in the presence of latent confounding. To correctly label every member of partitions $\mathbf{Z}_1$, $\mathbf{Z}_2$, $\mathbf{Z}_3$, and $\mathbf{Z}_5$, we assume the following sufficient (but not necessary) conditions: C1, C2, C3, and C4.

C3 The absence of inter-partition active paths (Definition 3.2).
C4 Causal sufficiency in $\mathcal{G}_{XY\mathbf{Z}}$.

Note that C3 and C4 are not needed for VAS discovery. Given C3, all of $\mathbf{Z}_2$ (if any exist) will be marginally dependent on $\mathbf{Z}_4$ and will be identifiable by LDP. The second statement of C2 is trivially satisfied when C3 is satisfied (as $\mathbf{Z}_5$ shares no active paths with $\mathbf{Z}_1$ is this setting) but is significant when C3 is violated. We demonstrate robustness of partition label correctness to specific violations of C3 in Tables G.6, G.10. Correctness under violations of C4 is described in Section 6.2, Appendix D.3, and Appendix D.2.

Given these sufficient (but not necessary) conditions, we obtain Theorem D.1.

**Theorem D.1** (Partition correctness of Algorithm 1). *Given the sufficient conditions described above, Algorithm 1 is guaranteed to output a correct partition of $\mathbf{Z}$ as defined in Table 1.*

Proof of Theorem D.1 follows from proofs of Lemmas D.2–D.9, which prove correctness for each step of Algorithm 1 sequentially. In footnotes, we acknowledge certain partitioning behaviors that occur when condition C3 is violated. However, these acknowledgements are non-exhaustive.

**Lemma D.2** (Step 1 of Algorithm 1). $X \perp\!\!\!\perp Z \wedge Y \perp\!\!\!\perp Z \iff Z \in \mathbf{Z}_8$.

*Proof. Step 1 of Algorithm 1 correctly identifies* $\mathbf{Z}_8$. This subset of $\mathbf{Z}$ is the most trivial to identify, as it is does not share an active path with either exposure nor outcome in $\mathcal{G}_{XY\mathbf{Z}}$. By definition, any $Z_8 \in \mathbf{Z}_8$ is marginally independent of $X$ and marginally independent of $Y$. Additionally, no candidate $Z \in \mathbf{Z} \setminus \mathbf{Z}_8$ is marginally independent of both $X$ and $Y$. Thus, any $Z \in \mathbf{Z}$ satisfying $X \perp\!\!\!\perp Z \wedge Y \perp\!\!\!\perp Z$ belongs to $\mathbf{Z}_8$ and can be removed from further consideration. □

**Lemma D.3** (Step 2 of Algorithm 1). $X \perp\!\!\!\perp Z \wedge X \not\perp\!\!\!\perp Z|Y \iff Z \in \mathbf{Z}_4$.

*Proof. Step 2 of Algorithm 1 correctly identifies* $\mathbf{Z}_4$.[7] Variables in $\mathbf{Z}_4$ share an active path with outcome $Y$ in $\mathcal{G}_{XY\mathbf{Z}}$ but not exposure $X$. For any $Z_4 \in \mathbf{Z}_4$, this results in a $v$-structure $X \cdots \to Y \leftarrow \cdots Z_4$.[8] By definition, all such $v$-structures entail $X \perp\!\!\!\perp Z_4 \wedge X \not\perp\!\!\!\perp Z_4|Y$. Besides $\mathbf{Z}_4$, only $\mathbf{Z}_8$ is marginally independent of $X$. However, $\mathbf{Z}_8$ is not conditionally dependent on $X$ given $Y$. Thus, no subset of $\mathbf{Z}$ entails $X \perp\!\!\!\perp Z \wedge X \not\perp\!\!\!\perp Z|Y$ except $\mathbf{Z}_4$. Any variable passing the test in Step 2 is unambiguously a member of $\mathbf{Z}_4$. Further, $\mathbf{Z}_4$ is correctly identified for downstream use in Step 4 to identify $\mathbf{Z}_{\text{POST}}$. □

**Lemma D.4** (Step 3 of Algorithm 1). $Y \not\perp\!\!\!\perp Z \wedge Y \perp\!\!\!\perp Z|X \iff Z \in \mathbf{Z}_{5,7}$.

*Proof. Step 3 of Algorithm 1 correctly identifies* $\mathbf{Z}_{5,7}$. We prove both directions of the bidirectional statement by direct proof. This test will be passed under two conditions: 1) $Z \in \mathbf{Z}_7$ for any arbitrary $\mathcal{G}_{XY\mathbf{Z}}$ and 2) $Z \in \mathbf{Z}_5$ when $\mathcal{G}_{XY\mathbf{Z}}$ when $\mathbf{Z}_1$ is the empty set (i.e., there are no backdoor paths for $X$ and $Y$). Thus, this test will capture all $\mathbf{Z}_7$ under any circumstances but will additionally capture $\mathbf{Z}_5$ only when $\mathcal{G}_{XY\mathbf{Z}}$ is structured such that exposure $X$ blocks all backdoor paths from $\mathbf{Z}_5$ to outcome $Y$. Further, no subset of $\mathbf{Z}$ will pass the test in Step 3 but $\mathbf{Z}_{5,7}$. Partitions $\mathbf{Z}_1$, $\mathbf{Z}_2$, $\mathbf{Z}_3$, $\mathbf{Z}_4$, and $\mathbf{Z}_6$ are parents or effects of $Y$ and thus $X$ cannot block the flow of association between these partitions and $Y$. $\mathbf{Z}_8$ will not pass this test either, as it is not marginally dependent on $Y$. Therefore, $Y \not\perp\!\!\!\perp Z \wedge Y \perp\!\!\!\perp Z|X$ if and only if $Z$ is in $\mathbf{Z}_{5,7}$. Further, if $\mathbf{Z}_1$ is nonempty when LDP terminates, it can be concluded that variables passing this test are only $\mathbf{Z}_7$ (Line 24, Algorithm 1). □

---

[7] See Appendix D.3 for proof of the identifiability of $\mathbf{Z}_4$ under latent confounding.

[8] Note that this requires $X$ and $Y$ to be marginally dependent, an assumption made in Section 4.1. $X \not\perp\!\!\!\perp Y$ is true when at least one of the following conditions is true: 1) $X$ is a direct cause of $Y$, 2) $X$ is an indirect cause of $Y$ through mediators in $\mathbf{Z}_3$, and/or 3) $X$ and $Y$ share confounders in $\mathbf{Z}_1$.

**Lemma D.5** (Step 4 of Algorithm 1). *Given execution of prior steps in Algorithm 1, $\exists Z_4 \in \mathbf{Z}_4 : Z \not\perp\!\!\!\perp Z_4$ or $Z \perp\!\!\!\perp Z_4 | X \cup Y \iff Z \in \mathbf{Z}_{2,3,6} \in \mathbf{Z}_{\text{POST}}$.*

*Proof.* Step 4 of Algorithm 1 correctly identifies $\mathbf{Z}_{2,3,6} \in \mathbf{Z}_{\text{POST}}$. This test exploits prior knowledge of $\mathbf{Z}_4$ to identify all of $\mathbf{Z}_2$ and $\mathbf{Z}_6$ in any arbitrary $\mathcal{G}_{XY\mathbf{Z}}$ meeting sufficient conditions C3–C4. Under condition C3, no $\mathbf{Z}_3$ will pass this test by the same logic that $\{\mathbf{Z}_1, \mathbf{Z}_5\}$ will not (as proven below).[9] Note that $\mathbf{Z}_4$, $\mathbf{Z}_7$, and $\mathbf{Z}_8$ have already been identified and removed from further consideration. Thus, this test must correctly identify $\mathbf{Z}_2$ and $\mathbf{Z}_6$ and must not incorrectly label these partitions as $\mathbf{Z}_1$ or $\mathbf{Z}_5$. We demonstrate correctness by direct proof of both directions of the bidirectional statement.

Under the assumption that $X$ and $Y$ are marginally dependent (Section 4.1), any $Z \in \mathbf{Z}_1 \cup \mathbf{Z}_5$ will form a $v$-structure $Z_4 \cdots \to Y \leftarrow \cdots Z$, but members of $\mathbf{Z}_2 \cup \mathbf{Z}_6$ will not (Figure 3). Such a $v$-structure implies that $Z \perp\!\!\!\perp Z_4$ and $Z \not\perp\!\!\!\perp Z_4 | X \cup Y$. As we seek to identify candidates $Z$ that do not induce such a $v$-structure, we logically negate these independence statements to test for $\mathbf{Z}_2$ and $\mathbf{Z}_6$. According to De Morgan's Laws, the negation of a conjunction is the disjunction of the negations. This yields the logical equivalence

$$\neg\left[(Z \perp\!\!\!\perp Z_4) \wedge (Z \not\perp\!\!\!\perp Z_4 | X \cup Y)\right] \equiv (Z \not\perp\!\!\!\perp Z_4) \vee (Z \perp\!\!\!\perp Z_4 | X \cup Y). \qquad \text{Per De Morgan's Laws.} \qquad (1)$$

Thus, when $Z \not\perp\!\!\!\perp Z_4$ or $Z \perp\!\!\!\perp Z_4 | X \cup Y$ is true, we will identify $\mathbf{Z}_2 \cup \mathbf{Z}_6$ but not $\mathbf{Z}_1 \cup \mathbf{Z}_5$. Likewise, when $Z \in \mathbf{Z}_2 \cup \mathbf{Z}_6$, a $v$-structure $Z_4 \to Y \leftarrow Z$ will never arise and thus $Z \not\perp\!\!\!\perp Z_4$ or $Z \perp\!\!\!\perp Z_4 | X \cup Y$. $\qquad \square$

To support Lemmas D.7-D.9, we introduce Proposition D.6.

**Proposition D.6.** *For any $Z_1 \in \mathbf{Z}_1$ that has an indirect active path to outcome $Y$, there must exist another $Z_1$ that is directly adjacent to $Y$. This extends analogously to indirect active paths between $\mathbf{Z}_1$ and $X$.*

**Lemma D.7** (Step 5 of Algorithm 1). *Given execution of prior steps in Algorithm 1, if $Y \not\perp\!\!\!\perp Z \wedge Y \perp\!\!\!\perp Z | X \cup \mathbf{Z}' \setminus Z$ then $Z \in \mathbf{Z}_{1,2,3,5} \in \mathbf{Z}_{\text{MIX}}$, and all backdoor paths between $\mathbf{Z}_{\text{MIX}}$ and $Y$ are blocked by $X$ and the members of $\mathbf{Z}$ that have not yet been labeled.*

*Proof.* Step 5 of Algorithm 1 correctly identifies $\mathbf{Z}_{\text{MIX}}$. Here, will assume that $\mathbf{Z}_5$ was not yet discovered at Step 3. We will prove that the conditioning set used in Step 5 correctly blocks all backdoor paths between $\mathbf{Z}_{\text{MIX}}$ and $Y$. Given sufficient conditions C1–C4, $\mathbf{Z}_2$, $\mathbf{Z}_4$, $\mathbf{Z}_6$, and $\mathbf{Z}_8$ have been previously identified and removed from further consideration.[10] Thus, we assume that only $\mathbf{Z}_1$, $\mathbf{Z}_3$, and $\mathbf{Z}_5$ are remaining in $\mathbf{Z}'$. By conditioning on $X \cup \mathbf{Z}' \setminus Z$, backdoor paths for $\{X, Y\}$ are blocked due to the inclusion of all $\mathbf{Z}_1 \in \mathbf{Z}'$. Thus, conditioning on $X \cup \mathbf{Z}' \setminus Z$ blocks all causal and non-causal association between $Z$ and $Y$. For all $Z \in \mathbf{Z}_5$, $Y \perp\!\!\!\perp Z | X \cup \mathbf{Z}' \setminus Z$. For any $Z \in \mathbf{Z}_1$ or $Z \in \mathbf{Z}_3$ that is not directly adjacent to $Y$, $Y \perp\!\!\!\perp Z | X \cup \mathbf{Z}' \setminus Z$. All members of $\mathbf{Z}_1$ and $\mathbf{Z}_3$ that are adjacent to $Y$ will proceed to be identified at Step 6. Thus, $\mathbf{Z}_{\text{MIX}}$ will consist of $\mathbf{Z}_5$, a fraction of $\mathbf{Z}_1$ (which may be the empty set), and a fraction of $\mathbf{Z}_3$ (which may be the empty set). $\qquad \square$

**Lemma D.8** (Step 6 of Algorithm 1). *Let $\mathbf{Z}_{\text{MIX}} = \mathbf{Z}_{\text{MIX}} \cup \mathbf{Z}_{5,7}$. Given execution of prior steps in Algorithm 1, if $\exists Z_{\text{MIX}} \in \mathbf{Z}_{\text{MIX}}$ such that $Z_{\text{MIX}} \perp\!\!\!\perp Z$ and $Z_{\text{MIX}} \not\perp\!\!\!\perp Z | X$ then $Z \in \mathbf{Z}_1$ and $Z_{\text{MIX}} \in \mathbf{Z}_{1,5}$. Else, $Z \in \mathbf{Z}_{\text{POST}}$. After execution of these tests, we loop through the remaining $\mathbf{Z}_{\text{MIX}}$ again. If $\exists Z_{1,5} \in \mathbf{Z}_{1,5}$ such that $Z_{1,5} \perp\!\!\!\perp Z_{\text{MIX}}$ and $Z_{1,5} \not\perp\!\!\!\perp Z_{\text{MIX}} | X$, then $Z_{\text{MIX}} \in \mathbf{Z}_1$. Else, $Z_{\text{MIX}} \in \mathbf{Z}_{\text{POST}}$.*

*Proof.* Step 6 of Algorithm 1 correctly differentiates $\mathbf{Z}_1$, $\mathbf{Z}_{1,5}$, $\mathbf{Z}_7$, and $\mathbf{Z}_{\text{POST}}$. This step relies on prior knowledge of $\mathbf{Z}_{\text{MIX}}$, which is gained programmatically through Steps 3 and 5. Under sufficient conditions C1–C4, $\mathbf{Z}_{\text{MIX}}$ initially contains $\mathbf{Z}_5$ and the members of $\mathbf{Z}_1$ and $\mathbf{Z}_3$ that are not adjacent to $Y$. At Step 6, we begin by unioning $\mathbf{Z}_{\text{MIX}}$ with $\mathbf{Z}_{5,7}$ as a safeguard in case any member of $\mathbf{Z}_5$ was lumped with $\mathbf{Z}_7$ at Step 3.

Step 6 exploits the presence of $v$-structures $Z \cdots \to X \leftarrow \cdots Z_1$ in $\mathcal{G}_{XY\mathbf{Z}}$. For any $\mathcal{G}_{XY\mathbf{Z}}$ (even when sufficient conditions are not met), the variables that can form such a $v$-structure with a $Z_1 \in \mathbf{Z}_1$ are 1) a $Z_5 \in \mathbf{Z}_5$ or 2) another $Z_1 \in \mathbf{Z}_1$ that does not share an active path with the first.

---

[9] If sufficient condition C3 is violated, a $Z_3$ may be captured at this step if it is marginally dependent on any $Z_4$. Further, this violation can cause Step 4 to miss members of $\mathbf{Z}_2$ that are not descendants of $Y$ (as discussed throughout Section D.2).

[10] If sufficient condition C3 is violated, members of $\mathbf{Z}_2$ that were not marginally dependent on any $Z_4 \in \mathbf{Z}_4$ (and thus not identified at Step 4) could be placed in $\mathbf{Z}_{\text{MIX}}$ at Step 5 instead. We prove in Section D.2 that the presence of $\mathbf{Z}_2$ in $\mathbf{Z}_{\text{MIX}}$ does not undermine the validity of the adjustment set returned by Algorithm 1.

First, we prove the first phase of Step 6. Under sufficient condition C3, $\mathbf{Z}_5 \cdots \to X \leftarrow \cdots \mathbf{Z}_1$ for all $\{\mathbf{Z}_1, \mathbf{Z}_5\}$. This means that all of $\mathbf{Z}_5$ is marginally independent of $\mathbf{Z}_1$, but is conditionally dependent on $\mathbf{Z}_1$ given $X$. As described in sufficient condition C2, the existence of at least two non-overlapping backdoor paths in $\mathcal{G}_{XY\mathbf{Z}}$ can also enable some $Z_1$ to form a $v$-structure at $X$ with another member of $\mathbf{Z}_1$. Thus, when a $v$-structure $Z_{\mathrm{MIX}} \cdots \to X \leftarrow \cdots Z$ is detected, then $Z$ must be in $\mathbf{Z}_1$ and $Z_{\mathrm{MIX}}$ must be in $\mathbf{Z}_{1,5}$. By extension, $Z_{\mathrm{MIX}}$ is not in $\mathbf{Z}_{\mathrm{POST}}$ nor $\mathbf{Z}_7$, and can be removed from the latter if it had been placed there at Step 3. Else, $Z$ must be in $\mathbf{Z}_{\mathrm{POST}}$.

Finally, we prove the second phase of Step 6. Variables still in $\mathbf{Z}_{\mathrm{MIX}}$ must be tested to distinguish the remaining members in $\mathbf{Z}_1$ from those in $\mathbf{Z}_{\mathrm{POST}}$. Any ground truth member of $\mathbf{Z}_1$ that remains in $\mathbf{Z}_{\mathrm{MIX}}$ at this point must be marginally dependent on all previously discovered members of $\mathbf{Z}_1$, otherwise these would have already been placed in $\mathbf{Z}_{1,5}$. By this point, all of $\mathbf{Z}_5$ is now contained in $\mathbf{Z}_{1,5}$. Under sufficient condition C3, $\mathbf{Z}_1 \perp\!\!\!\perp \mathbf{Z}_5$ but $\mathbf{Z}_{\mathrm{POST}} \not\!\perp\!\!\!\perp \mathbf{Z}_5$. Thus, testing $\mathbf{Z}_{\mathrm{MIX}}$ against $\mathbf{Z}_{1,5}$ for marginal independence will differentiate the remaining $\mathbf{Z}_1 \in \mathbf{Z}_{\mathrm{MIX}}$ from the remaining $\mathbf{Z}_{\mathrm{POST}} \in \mathbf{Z}_{\mathrm{MIX}}$. $\qquad\square$

**Lemma D.9** (Step 7 of Algorithm 1). *Given execution of prior steps in Algorithm 1, if $\exists\, Z_1 \in \mathbf{Z}_1$ and $Z_{1,5} \in \mathbf{Z}_{1,5}$ such that $Z_{1,5} \not\!\perp\!\!\!\perp Z_1$, then $Z_{1,5} \in \mathbf{Z}_1$. Else, $Z_{1,5} \in \mathbf{Z}_5$.*

*Proof.* Step 7 of Algorithm 1 correctly differentiates $\mathbf{Z}_1$ from $\mathbf{Z}_5$. This step handles cases exemplified by node $B_1$ in the butterfly structure of Figure E.1, which can have arbitrarily long, indirect, yet active paths to $Y$. During Step 5, the conditioning set $X \cup \mathbf{Z}' \setminus Z$ contains all $\mathbf{Z}_1$, among other variables. For a $B_1$-type confounder, this conditioning set blocks all backdoor paths to $Y$, triggering the test to label the node as a member of $\mathbf{Z}_{\mathrm{MIX}}$. To detect such a case, observe that $B_1$-type confounders have marginal dependence on the subset of $\mathbf{Z}_1$ that was discovered at Step 6. All $Z_1 \in \mathbf{Z}_1$ previously discovered at Step 6 are directly adjacent to $Y$. Under sufficient condition C3, all of $\mathbf{Z}_5$ is marginally independent of $\mathbf{Z}_1$. Even when sufficient condition C3 is violated, no member of $\mathbf{Z}_5$ will ever be dependent on a $Z_1$ that is directly adjacent to $Y$. Therefore, any member of $\mathbf{Z}_{1,5}$ that is marginally dependent on at least one member of $\mathbf{Z}_1$ discovered at Step 6 must be in $\mathbf{Z}_1$. If such marginal dependence is not detected between a given $Z_{1,5}$ and any member of $\mathbf{Z}_1$ discovered at Step 6, then $Z_{1,5} \in \mathbf{Z}_5$ instead. $\qquad\square$

## D.2   VAS UNDER LATENT CONFOUNDING

Here we prove Theorem 4.5, which states that LDP returns VAS when Conditions C1, C2, and the $\mathbf{Z}_5$ criterion (Definition 4.3) are satisfied.

Recall the definition of a VAS under the backdoor criterion (Definition 2.4). Let $\mathbf{A}_{XY}$ be an adjustment set for $\{X, Y\}$ that does not contain $\{X, Y\}$. We say that $\mathbf{A}_{XY}$ is valid if

*Item 1* $\mathbf{A}_{XY}$ contains no descendants of $X$; and
*Item 2* $\mathbf{A}_{XY}$ blocks all backdoor paths for $X$ and $Y$ [Pearl, 2009].

The set $\mathbf{A}_{XY}$ returned by LDP is synonymous with partition $\mathbf{Z}_1$. To prove Theorem 4.5, we must prove that both *Item 1* and *Item 2* always hold for the $\mathbf{Z}_1$ returned by Algorithm 1 under sufficient conditions.

To prove *Item 1*, we prove Lemma 4.2 with the help of Proposition D.10.

**Proposition D.10.** *If two variables are marginally or conditionally dependent and the conditioning set remains unchanged, the addition of a new active path in $\mathcal{G}_{XY\mathbf{Z}}$ cannot render them independent.*

*Proof.* Intuition for proof of Lemma 4.2 follows from the fact that all descendants of $X$ are marginally dependent on all of $\mathbf{Z}_1$ and all of $\mathbf{Z}_5$ by definition, and will be placed in $\mathbf{Z}_{\mathrm{POST}}$ by Step 6 or earlier. This marginal dependence is detectable even if the descendants of $X$ are latently confounded, and cannot be negated by inter-partition active paths (Proposition D.10). As any causal path from $X$ to $Y$ features an edge out of $X$ ($X \to \cdots$), this also guarantees that no causal path from $X$ to $Y$ will be blocked by the $\mathbf{Z}_1$ returned by LDP. $\qquad\square$

To prove *Item 2*, we prove Lemma 4.4 and show that the $\mathbf{Z}_5$ criterion is a valid indicator that backdoor paths are blocked by the recovered $\mathbf{Z}_1$.

*Proof.* This lemma states that the discovery of at least one $Z_5 \in \mathbf{Z}_5$ that is $d$-separable from $Y$ given $X \cup \mathbf{Z}_1$ indicates that all backdoor paths for $\{X, Y\}$ are blocked. Recall the definition of $\mathbf{Z}_5$: non-descendants of $X$ whose causal effects on $Y$ are fully mediated by $X$, and that share no confounders with $Y$. If $\mathbf{Z}_1$ is the empty set, then $\mathbf{Z}_5$ is definitionally $d$-separable

from $Y$ given $X$. When $\mathbf{Z}_1$ is non-empty, conditioning on $X$ opens backdoor paths for $\{X, Y\}$ through $\mathbf{Z}_1$. These backdoor paths can only be reblocked by adjusting for a sufficient subset of $\mathbf{Z}_1$. If the $\mathbf{Z}_1$ observed in $\mathbf{Z}$ do not block all backdoor paths for $X$ and $Y$ in the true underlying graph, then $\mathbf{Z}_5$ will not be $d$-separable from $Y$ given $X$ and the recovered $\mathbf{Z}_1$. Thus, the $\mathbf{Z}_5$ criterion will be passed only if the recovered $\mathbf{Z}_1$ block all backdoor paths for $X$ and $Y$. $\qquad\square$

## D.3   PARTITION CORRECTNESS IN ARBITRARY DAGS UNDER LATENT CONFOUNDING

Here we prove Theorem 4.6, which implies that the identification of $\mathbf{Z}_4$, $\mathbf{Z}_7$, and $\mathbf{Z}_8$ does not require Conditions C1–C4.

*Proof.*  As shown in Lemmas D.2, D.3, and D.4, the following bidirectional statements define $\mathbf{Z}_4$, $\mathbf{Z}_7$, and $\mathbf{Z}_8$.

$$X \perp\!\!\!\perp Z \wedge Y \perp\!\!\!\perp Z \iff Z \in \mathbf{Z}_8 \qquad \text{Definition of causal partition } \mathbf{Z}_8 \text{ (Step 1).}$$
$$X \perp\!\!\!\perp Z \wedge X \not\perp\!\!\!\perp Z|Y \iff Z \in \mathbf{Z}_4 \qquad \text{Definition of causal partition } \mathbf{Z}_4 \text{ (Step 2).}$$
$$Y \not\perp\!\!\!\perp Z \wedge Y \perp\!\!\!\perp Z|X \iff Z \in \mathbf{Z}_{5,7} \qquad \text{Definition of } \mathbf{Z}_7 \text{ in all structures and } \mathbf{Z}_5 \text{ when } |\mathbf{Z}_1| = 0 \text{ (Step 3).}$$

These statements are fundamental properties of $\mathbf{Z}_4$, $\mathbf{Z}_7$, and $\mathbf{Z}_8$ in all DAGs, irrespective of the inter-partition active paths in which they participate. For $\mathbf{Z}_5$, the conditional independence relations in Step 3 define this partition only when $\mathbf{Z}_1$ is the empty set. Thus, these statements both precisely define partitions $\mathbf{Z}_4$, $\mathbf{Z}_7$, and $\mathbf{Z}_8$ in arbitrary structures and are also the statistical tests used to detect them. Further, these tests rely only on knowledge of $\{X, Y, Z\}$, and no other variables. Thus, failure to observe variables other than $Z$ in the ground truth graph have no impact on the outcome of these tests. This is in contrast to Step 5, for example, which relies on access to additional variables in the true graph for correctness. Thus, identification of $\mathbf{Z}_4$, $\mathbf{Z}_7$, and $\mathbf{Z}_8$ is robust to latent confounding and inter-partition active paths, and relies only on knowledge of $\{X, Y, Z\}$ in arbitrary structures. These facts render C1–C4 unnecessary. Note that when $\mathbf{Z}_1$ is the empty set in the true graph, $\mathbf{Z}_5$ and $\mathbf{Z}_7$ are not differentiated by LDP and remain in the same superset. Nevertheless, this superset is guaranteed to be correctly labeled. $\qquad\square$

# E GRAPHS FOR EXPERIMENTAL VALIDATION

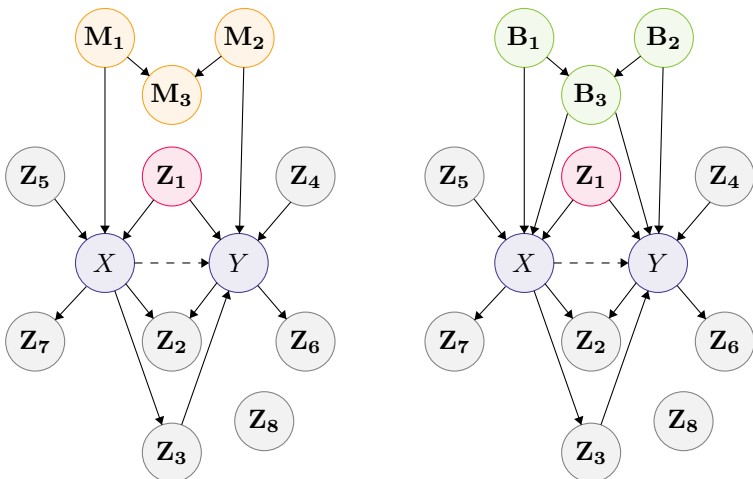

Figure E.1: M-structures and butterfly structures [Ding and Miratrix, 2014]. Ten-node DAG plus M-structure (left) and ten-node DAG plus butterfly structure (right). Note that $M_1 \in Z_5$, $M_2 \in Z_4$, $M_3 \in Z_2$, and $\{B_1, B_2, B_3\} \in Z_1$. Performance of LDP on these structures is reported in Table G.6.

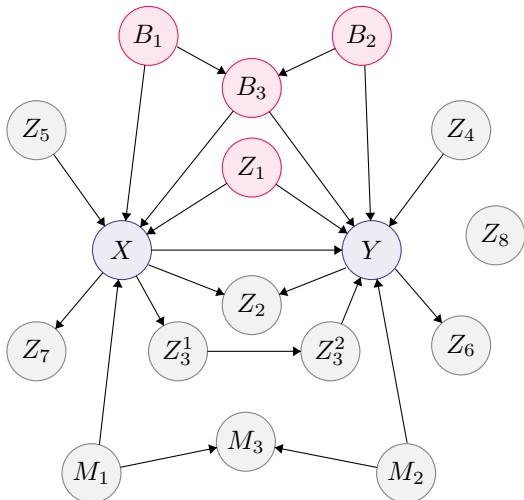

Figure E.2: Seventeen-node DAG with M-structure, butterfly structure, and mediator chain. Note that $M_1 \in Z_5$, $M_2 \in Z_4$, $M_3 \in Z_2$, and $\{B_1, B_2, B_3\} \in Z_1$. Nodes highlighted in red ($\{Z_1, B_1, B_2, B_3\}$) represent all confounders for $\{X, Y\}$. Performance of LDP on this structure is reported in Table G.7.

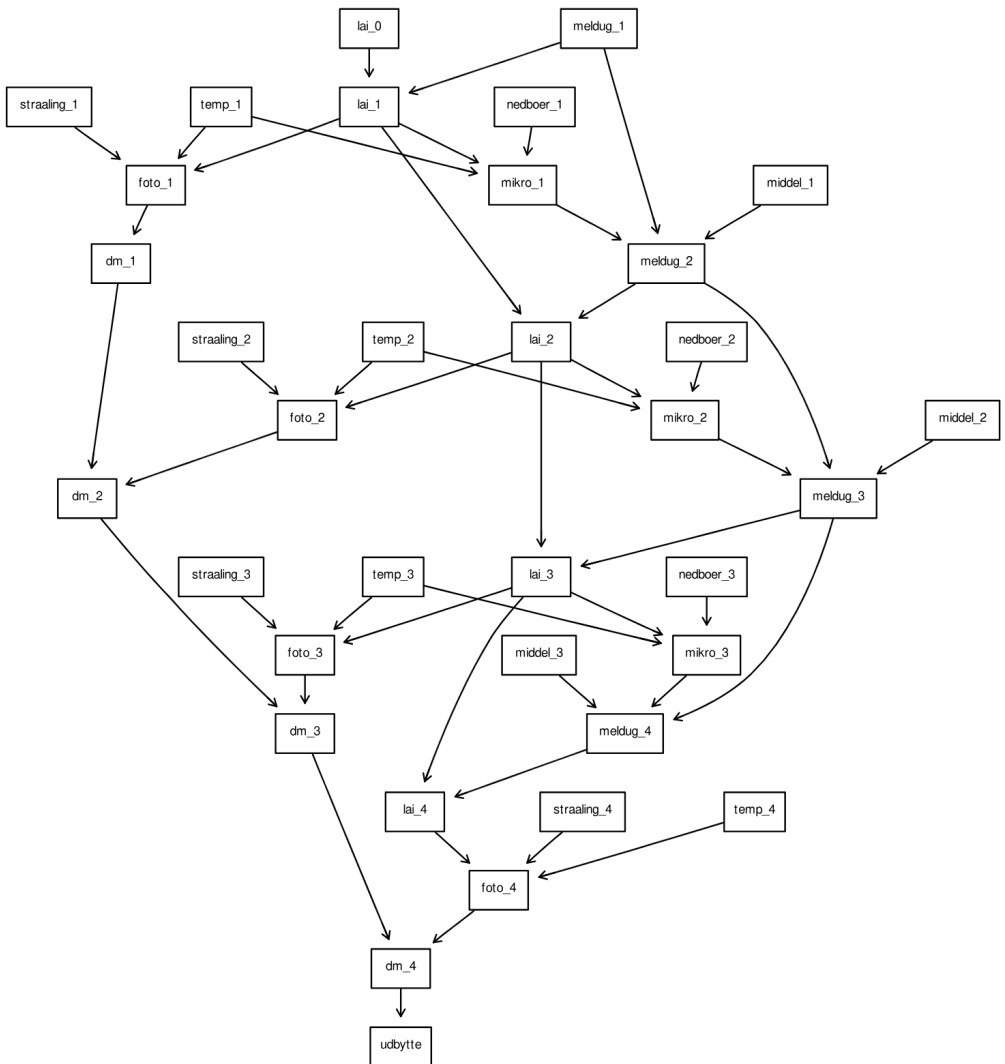

Figure E.3: The complete ground truth MILDEW DAG [Jensen and Jensen, 1996] obtained from `bnlearn` [Scutari, 2010]. The ground truth DAG contains 35 nodes, 46 edges, and 540150 parameters. The average Markov blanket size, average degree, and maximum in-degree are 4.57, 2.63, and 3, respectively. Inference and evaluation omit variables DM_1 and FOTO_1 due to independence test challenges with LDP, MB-by-MB, and LDECC, including those described in Section G.1 for MB-by-MB and LDECC (which were made more severe by inclusion of these nodes). Performance on this structure is reported in Figure 6 an Table G.8.

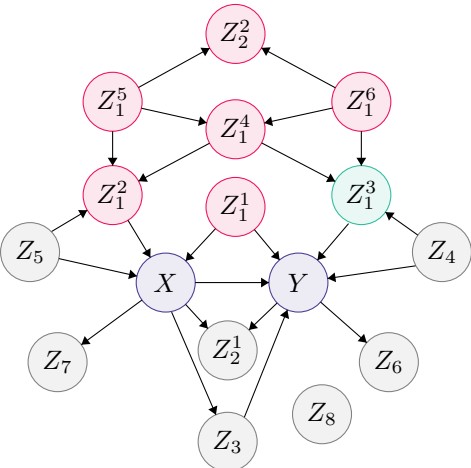

Figure E.4: A complex backdoor path illustrates a known failure mode of LDP partition labeling that is still successful for valid adjustment set identification. In theory, all nodes highlighted in red will be placed in $\mathbf{Z}_1$ by LDP. Even though $Z_1^2$ is adjacent to the only instrument in this DAG, this confounder will be discoverable due to its marginal independence with $Z_1^1$. Due to its marginal dependence on $Z_4$, confounder $Z_1^3$ will be mislabeled and placed in $\mathbf{Z}_{\text{POST}}$ by LDP. This mislabeling persists even under infinite data. Due to its marginal independence with $Z_4$, collider $Z_2^2$ will be mislabeled and placed in $\mathbf{Z}_1$. Despite these mislabelings, the red node set constitutes a valid adjustment set. LDP returned a valid adjustment set for this structure for 99% (99/100) of replicates at $n = 5k$ samples and 98% (98/100) of replicates at $n = 10k$ samples. Noise was hypergeometric and causal mechanisms were quadratic (chi-square independence test; $\alpha = 0.001$).

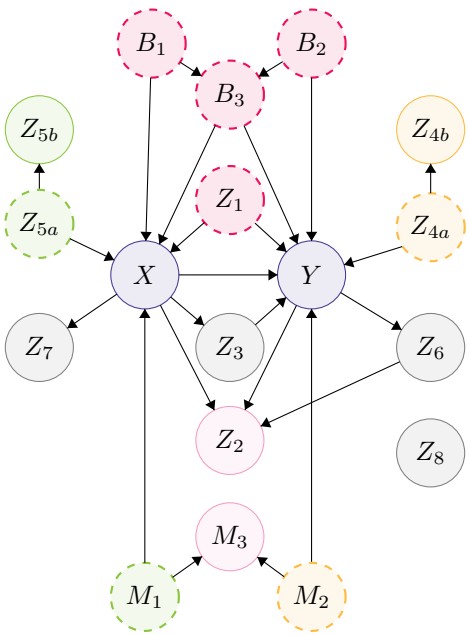

Figure E.5: An 18-node DAG with a butterfly structure, M-structure, and inter-partition active paths. Members of $\mathbf{Z}_1$ are pictured in red, $\mathbf{Z}_2$ in pink, $\mathbf{Z}_4$ in yellow, and $\mathbf{Z}_5$ in green. This structure was used to experimentally demonstrate the robustness of LDP to latent confounding, where one to three nodes with a dashed perimeter were dropped per iteration. Results are reported in Section 6 and Table G.10.

# F  EXPERIMENTAL DESIGN

## F.1  EXPERIMENTAL DATA

**Synthetic DAGs**   Theoretical guarantees were validated for four causally sufficient DAG structures (Figures 3, E.1, E.2, E.4) and one structure with hidden variables (Figure E.5). In the discrete data simulations, we used 12 data generating processes for the 10-node DAG (Figure 3), four processes for both 13-node DAGs (Figure E.1), and two processes for the 17-node DAG (Figure E.2). Causal mechanisms were linear and nonlinear. Six linear-continuous data generating processes were simulated for the 10-node DAG (Figure 3).

**MILDEW Benchmark**   The MILDEW network models fungicide use against powdery mildew in winter wheat [Jensen and Jensen, 1996]. We selected one exposure-outcome pair (MIKRO_1 → MELDUG_2) that meets sufficient conditions for LDP. All variables are categorical. $\mathbf{Z}$ contains 31 nodes in $\{\mathbf{Z}_1, \mathbf{Z}_2, \mathbf{Z}_4, \mathbf{Z}_5, \mathbf{Z}_8\}$, with a low proportion of confounders ($|\mathbf{Z}_1| = 2$) and high proportion of colliders ($|\mathbf{Z}_2| = 14$). Data were sampled using the `bnlearn` R package [Scutari, 2010]. Figure E.3 further describes the DAG used for inference and evaluation.

## F.2  BASELINE METHODS

**PC Algorithm**   PC is a classic global causal discovery algorithm that provides asymptotic theoretical guarantees [Spirtes et al., 2000]. It assumes causal Markov, faithfulness, and causal sufficiency and returns a MEC. The worst-case time complexity for PC is exponential in the number of nodes, as demonstrated in Figure 5. Experiments use the implementation by Kalisch and Buhlmann [2007][11], unless otherwise noted.

**MB-by-MB**   MB-by-MB [Wang et al., 2014] infers the local structure around a target node to distinguish parents from children. It sequentially learns Markov blankets (MBs) and the local structures within these, starting from the target node, moving to its neighbors, and so on. It terminates when the parents and children of the target are discovered or if it is not possible to distinguish them, returning the induced *completed partially directed acyclic graph* (CPDAG) over the target and its neighbors. Experiments use an implementation that combines IAMB [Tsamardinos et al., 2003, Fig. 2] and PC for every sequential step. Like PC, time complexity is worst-case exponential in node count.

**Local Discovery using Eager Collider Checks (LDECC)**   LDECC [Gupta et al., 2023] is a local discovery algorithm that infers the induced CPDAG over a given target node and its neighbors. Unlike MB-by-MB, LDECC does not proceed sequentially and runs conditional independence tests in a similar order as PC, leveraging discovered unshielded colliders to immediately orient the edges around the target node. LDECC is provably polynomial-time for certain categories of DAGs, but exponential for others.

**Baseline Evaluation**   Let $\mathbf{A}_{XY}$ be any adjustment set for $\{X, Y\}$ returned by a method in this study. Let $\mathbf{A}_{CC} := \{\mathbf{Z}_1\}$ and $\mathbf{A}_{DC} := \{\mathbf{Z}_1, \mathbf{Z}_4, \mathbf{Z}_5\}$ be valid adjustment sets for $\{X, Y\}$ under the *common cause criterion* (CCC) and *disjunctive cause criterion* (DCC), respectively [VanderWeele and Shpitser, 2011]. For PC, $\mathbf{A}_{CC} := \text{ancestors}(X) \cap \text{ancestors}(Y) = \{\mathbf{Z}_1\}$, and $\mathbf{A}_{DC} := \{\text{ancestors}(X) \cup \text{ancestors}(Y) \setminus \text{descendants}(X)\} = \{\mathbf{Z}_1, \mathbf{Z}_4, \mathbf{Z}_5\}$, where ancestors and descendants hold for all members of the MEC. As MB-by-MB and LDECC only return the direct parents and children of a single target, we run these baselines with $X$ and $Y$ as separate targets and cache intermediate results to prevent redundant independence testing. $\mathbf{A}_{DC} := \{\text{parents}(X) \cup \text{parents}(Y) \setminus \text{children}(X)\} = \mathbf{Z}_1' \cup \mathbf{Z}_4 \cup \mathbf{Z}_5$, where $\mathbf{Z}_1'$ is directly adjacent to $X$, $Y$, or both (but not neither). $\mathbf{A}_{CC} := \{\text{parents}(X) \cap \text{parents}(Y)\}$, i.e., all confounders directly adjacent to both $X$ and $Y$. Thus, $\mathbf{A}_{CC}$ under LDECC and MB-by-MB are not guaranteed to block all backdoor paths.

---

[11]https://github.com/keiichishima/pcalg

# G EXPERIMENTAL RESULTS

| | | | MEAN RUNTIME (SECONDS) | | | | TESTS PER RUN | | | |
|---|---|---|---|---|---|---|---|---|---|---|
| $|\mathbf{Z}|$ | $|\mathbf{Z}_-|$ | LDP:$|\mathbf{Z}|^2$ | LDP | LDECC | MB-BY-MB | PC | LDP | LDECC | MB-BY-MB | PC |
| 8 | 1 | 0.781 | 0.0143 (0.0121-0.0165) | 0.1144 (0.1098-0.119) | 0.1205 (0.1163-0.1247) | 0.076 (0.0734-0.0787) | 50 | 641 | 513.6 (513.2-514.1) | 508 |
| 16 | 2 | 0.500 | 0.0299 (0.0287-0.0311) | 7.011 (6.7783-7.2437) | 9.3711 (9.0831-9.6591) | 8.7598 (8.5028-9.0169) | 128 | 23344.3 (23331.9-23356.7) | 29687.5 (29675.4-29699.5) | 29556 |
| 24 | 3 | 0.406 | 0.0410 (0.0399-0.0421) | - | - | - | 234 | - | - | - |
| 32 | 4 | 0.359 | 0.0587 (0.0574-0.0600) | - | - | - | 368 | - | - | - |
| 40 | 5 | 0.331 | 0.0838 (0.0827-0.0849) | - | - | - | 530 | - | - | - |
| 48 | 6 | 0.313 | 0.1230 (0.1215-0.1245) | - | - | - | 720 | - | - | - |
| 56 | 7 | 0.299 | 0.1492 (0.1476-0.1509) | - | - | - | 938 | - | - | - |
| 64 | 8 | 0.289 | 0.2016 (0.1963-0.2070) | - | - | - | 1184 | - | - | - |
| 72 | 9 | 0.281 | 0.2495 (0.2458-0.2533) | - | - | - | 1458 | - | - | - |
| 80 | 10 | 0.275 | 0.2836 (0.2784-0.2887) | - | - | - | 1760 | - | - | - |

Table G.1: Mean runtime and total independence tests performed per DAG as cardinality of $\mathbf{Z}$ ($|\mathbf{Z}|$) increases. Values are averaged over 100 replicates for DAGs analogous to Figure 3 (sample size $n = 1k$ each), with 95% confidence intervals in parentheses. All data generating processes feature hypergeometric noise with quadratic causal mechanisms (structural equation in Table G.2). Independence was determined by an oracle. Cardinality of each partition is reported as $|\mathbf{Z}_-|$. The ratio of true total tests for LDP to expected quadratic count is reported as LDP:$|\mathbf{Z}|^2$. Baselines were only evaluated up to $|\mathbf{Z}| = 16$ due to very high test counts. All experiments were run on a 2017 MacBook with 2.9 GHz Quad-Core Intel Core i7. Growth curves are plotted in Figure 5.

| DAG STRUCTURE | CAUSAL MECHANISM | NOISE DISTRIBUTION | $X \to Y$ | STRUCTURAL EQUATION |
|---|---|---|---|---|
| 10-node (Figure 3) | Linear | Bernoulli | True | $V_i = \lfloor(0.3 * sum(\mathbf{Pa}_i))\rfloor + \epsilon_i$ |
| 10-node (Figure 3) | Linear | Bernoulli | False | $V_i = \lfloor(0.45 * sum(\mathbf{Pa}_i))\rfloor + \epsilon_i$ |
| 10-node (Figure 3) | Linear | Hypergeometric | True | $V_i = \lfloor(0.3 * sum(\mathbf{Pa}_i))\rfloor + \epsilon_i$ |
| 10-node (Figure 3) | Linear | Hypergeometric | False | $V_i = \lfloor(0.45 * sum(\mathbf{Pa}_i))\rfloor + \epsilon_i$ |
| 10-node (Figure 3) | Quadratic | Bernoulli | True | $V_i = \lfloor(-1.4 * sum(\mathbf{Pa}_i)^2)\rfloor + \epsilon_i$ |
| 10-node (Figure 3) | Quadratic | Bernoulli | False | $V_i = \lfloor(-1.4 * sum(\mathbf{Pa}_i)^2)\rfloor + \epsilon_i$ |
| 10-node (Figure 3) | Quadratic | Hypergeometric | True | $V_i = \lfloor(0.4 * sum(\mathbf{Pa}_i)^2)\rfloor + \epsilon_i$ |
| 10-node (Figure 3) | Quadratic | Hypergeometric | False | $V_i = \lfloor(0.4 * sum(\mathbf{Pa}_i)^2)\rfloor + \epsilon_i$ |
| 10-node (Figure 3) | Cube root | Bernoulli | True | $V_i = \lfloor(1.2 * \sqrt[3]{(\mathbf{Pa}_i)})\rfloor + \epsilon_i$ |
| 10-node (Figure 3) | Cube root | Bernoulli | False | $V_i = \lfloor(1.2 * \sqrt[3]{(\mathbf{Pa}_i)})\rfloor + \epsilon_i$ |
| 10-node (Figure 3) | Cube root | Hypergeometric | True | $V_i = \lfloor(0.7 * \sqrt[3]{(\mathbf{Pa}_i)})\rfloor + \epsilon_i$ |
| 10-node (Figure 3) | Cube root | Hypergeometric | False | $V_i = \lfloor(0.7 * \sqrt[3]{(\mathbf{Pa}_i)})\rfloor + \epsilon_i$ |
| 13-node with M (Figure E.1) | Linear | Bernoulli | True | $V_i = \lfloor(1.5 * sum(\mathbf{Pa}_i))\rfloor + \epsilon_i$ |
| 13-node with M (Figure E.1) | Quadratic | Hypergeometric | True | $V_i = \lfloor(1.5 * sum(\mathbf{Pa}_i)^2)\rfloor + \epsilon_i$ |
| 13-node with butterfly (Figure E.1) | Linear | Bernoulli | True | $V_i = \lfloor(1.9 * sum(\mathbf{Pa}_i))\rfloor + \epsilon_i$ |
| 13-node with butterfly (Figure E.1) | Quadratic | Hypergeometric | True | $V_i = \lfloor(2.8 * sum(\mathbf{Pa}_i)^2)\rfloor + \epsilon_i$ |
| 18-node with latent confounding (Figure E.5) | Linear | Bernoulli | True | $V_i = \lfloor(1.3 * sum(\mathbf{Pa}_i))\rfloor + \epsilon_i$ |

Table G.2: Structural equations for all discrete synthetic data generating processes. $V_i$ denotes a random variable, $\mathbf{Pa}_i$ denotes the set of its direct causal parents, and $\epsilon_i$ denotes the random noise term associated with it. Fixed coefficients range across structural equations ($[-1.4, 2.8]$) to simulate varying effect sizes.

| DAG STRUCTURE | EXPERIMENT | CAUSAL MECHANISM | NOISE DISTRIBUTION | $X \to Y$ | STRUCTURAL EQUATION |
|---|---|---|---|---|---|
| 10-node (Figure 3) | Figure G.1 | Linear | Gaussian | True | $V_i = \sum(r * \mathbf{Pa}_i) + \epsilon_i$ |
| 10-node (Figure 3) | Figure G.1 | Linear | Gaussian | False | $V_i = \sum(r * \mathbf{Pa}_i) + \epsilon_i$ |
| 10-node (Figure 3) | Figure G.1 | Linear | Uniform | True | $V_i = \sum(r * \mathbf{Pa}_i) + \epsilon_i$ |
| 10-node (Figure 3) | Figure G.1 | Linear | Uniform | False | $V_i = \sum(r * \mathbf{Pa}_i) + \epsilon_i$ |
| 10-node (Figure 3) | Figure G.1 | Linear | Exponential | True | $V_i = \sum(r * \mathbf{Pa}_i) + \epsilon_i$ |
| 10-node (Figure 3) | Figure G.1 | Linear | Exponential | False | $V_i = \sum(r * \mathbf{Pa}_i) + \epsilon_i$ |
| 10-node (Figure 3) | Figure 6 | Linear | Gaussian | True | $V_i = \sum(c * \mathbf{Pa}_i) + \epsilon_i$ |

Table G.3: Structural equations for all continuous synthetic data generating processes. $V_i$ denotes a random variable, $\mathbf{Pa}_i$ the set of its direct causal parents, and $\epsilon_i$ the random noise term. Coefficient $r$ is selected uniformly at random from the interval $[1.0, 3.0)$. For the experiments reported in Figure 6, coefficient $c$ is 1.0 for all parents except for $X$ when causal for $Y$, in which case $c = 2.75$. For this DAG, the total effect of $X$ on $Y$ is 3.75, as the direct effect is 2.75 and the indirect effect through $\mathbf{Z}_3$ is 1.0.

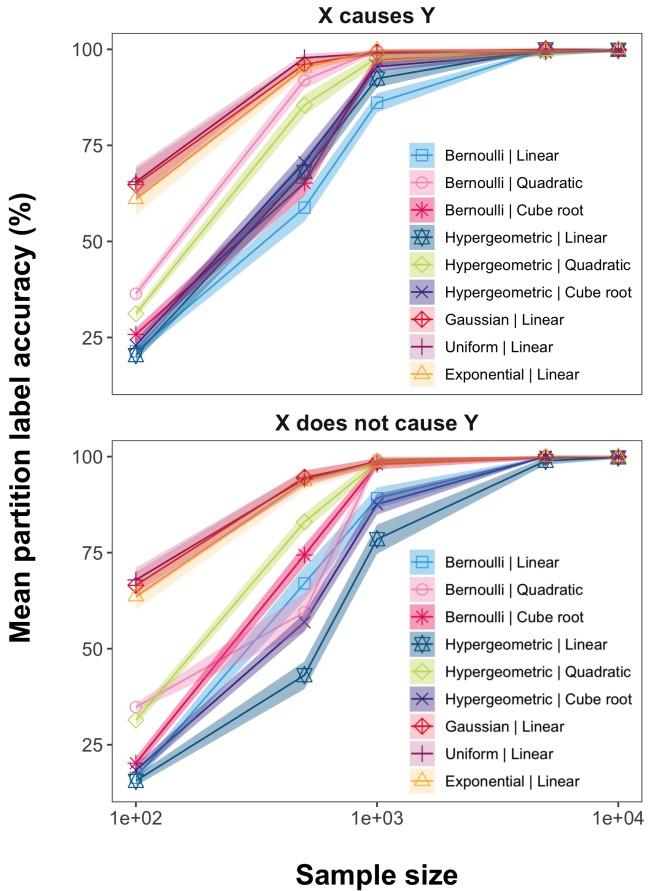

Figure G.1: Partition label accuracy of LDP on a 10-node DAG with one node per partition (Figure 3). Accuracy is averaged over 100 DAGs (i.e., 800 variables total, excluding exposure-outcome pairs), with 95% confidence intervals in shaded regions. Independence was determined by chi-square tests for discrete data and Fisher-z for continuous data, both with $\alpha = 0.001$. Tables G.4 and G.5 report raw data.

| 10-NODE GRAPH WITH BERNOULLI NOISE | | | | | |
|---|---|---|---|---|---|
| **LINEAR** | | **QUADRATIC** | | **CUBE ROOT** | |
| $X \to Y$ | $X \not\to Y$ | $X \to Y$ | $X \not\to Y$ | $X \to Y$ | $X \not\to Y$ |
| $n$ | | | | | |
| 100 | 21.4 (20.1-22.6) | 16.4 (15.2-17.5) | 36.4 (34.9-37.8) | 34.8 (33.7-35.8) | 25.8 (24.1-27.4) | 20.2 (18.7-21.8) |
| 500 | 58.8 (55.2-62.3) | 67.0 (62.7-71.3) | 92.0 (89.7-94.3) | 59.5 (54.5-64.5) | 65.2 (62.0-68.5) | 74.4 (71.6-77.1) |
| $1k$ | 86.1 (83.6-88.6) | 89.2 (86.5-92.0) | 99.8 (99.3-100) | 99.2 (98.2-100) | 97.2 (95.8-98.7) | 98.0 (96.7-99.3) |
| $5k$ | 99.9 (99.6-100) | 99.9 (99.6-100) | 100 (100-100) | 100 (100-100) | 99.9 (99.6-100) | 99.8 (99.3-100) |
| $10k$ | 100 (100-100) | 99.9 (99.6-100) | 100 (100-100) | 100 (100-100) | 99.6 (99.1-100) | 99.9 (99.6-100) |

| 10-NODE GRAPH WITH HYPERGEOMETRIC NOISE | | | | | |
|---|---|---|---|---|---|
| **LINEAR** | | **QUADRATIC** | | **CUBE ROOT** | |
| $X \to Y$ | $X \not\to Y$ | $X \to Y$ | $X \not\to Y$ | $X \to Y$ | $X \not\to Y$ |
| $n$ | | | | | |
| 100 | 20.4 (18.8-22.0) | 15.6 (14.3-17.0) | 31.2 (29.9-32.6) | 31.4 (29.9-32.9) | 23.0 (21.7-24.3) | 18.4 (16.9-19.8) |
| 500 | 68.1 (64.7-71.6) | 43.1 (39.5-46.7) | 85.5 (82.7-88.3) | 83.0 (80.7-85.3) | 70.6 (67.7-73.5) | 56.9 (54.5-59.2) |
| $1k$ | 92.4 (90.1-94.6) | 78.5 (74.7-82.3) | 97.8 (96.3-99.2) | 98.5 (97.2-99.8) | 95.5 (94.3-96.7) | 87.6 (84.9-90.3) |
| $5k$ | 100 (100-100) | 98.9 (97.9-99.8) | 99.2 (98.0-100) | 100 (100-100) | 99.2 (98.5-100) | 100 (100-100) |
| $10k$ | 99.9 (99.6-100) | 99.8 (99.4-100) | 99.8 (99.4-100) | 100 (100-100) | 99.8 (99.3-100) | 100 (100-100) |

Table G.4: Partition label accuracy of Algorithm 1 on a 10-node DAG (Figure 3) across discrete noise distributions, linear and nonlinar causal mechanisms, and sample sizes ($n$). All DAGs feature one node per partition ($\mathbf{Z}_1 - \mathbf{Z}_8$). Reported values are partition label accuracy averaged over 100 DAGs (i.e., 800 variables total, excluding all exposure-outcome pairs). The 95% confidence interval is reported in parentheses. Data generating processes where $X$ is a direct cause of $Y$ are denoted by $X \to Y$, with $X \not\to Y$ denoting no direct causal effect of $X$ on $Y$. Independence was determined by chi-square tests ($\alpha = 0.001$). All experiments were run on a 2017 MacBook with 2.9 GHz Quad-Core Intel Core i7.

| 10-NODE GRAPH WITH CONTINUOUS NOISE | | | | | |
|---|---|---|---|---|---|
| **GAUSSIAN │ LINEAR** | | **UNIFORM │ LINEAR** | | **EXPONENTIAL │ LINEAR** | |
| $X \to Y$ | $X \not\to Y$ | $X \to Y$ | $X \not\to Y$ | $X \to Y$ | $X \not\to Y$ |
| $n$ | | | | | |
| 100 | 64.8 (60.7-68.8) | 66.5 (63.0-70.0) | 65.5 (61.4-69.6) | 67.9 (64.6-71.1) | 61.0 (56.9-65.1) | 63.6 (60.2-67.0) |
| 500 | 96.1 (94.4-97.8) | 94.6 (92.8-96.5) | 97.8 (96.6-98.9) | 94.2 (92.2-96.3) | 95.2 (93.5-97.0) | 93.6 (91.3-95.9) |
| $1k$ | 99.2 (98.7-99.8) | 98.5 (97.5-99.5) | 99.0 (98.3-99.7) | 98.8 (97.8-99.7) | 99.8 (99.4-100) | 98.5 (97.4-99.6) |
| $5k$ | 100 (100-100) | 99.9 (99.6-100) | 99.6 (98.9-100) | 99.8 (99.3-100) | 99.5 (98.7-100) | 99.9 (99.6-100) |
| $10k$ | 99.9 (99.6-100) | 99.9 (99.6-100) | 100 (100-100) | 100 (100-100) | 100 (100-100) | 100 (100-100) |

Table G.5: Partition label accuracy of Algorithm 1 on a 10-node DAG (Figure 3) across continuous noise distributions, linear causal mechanisms, and sample sizes ($n$). All DAGs feature one node per partition ($\mathbf{Z}_1 - \mathbf{Z}_8$). Reported values are partition label accuracy averaged over 100 DAGs (i.e., 800 variables total, excluding all exposure-outcome pairs). The 95% confidence interval is reported in parentheses. Data generating processes where $X$ is a direct cause of $Y$ are denoted by $X \to Y$, with $X \not\to Y$ denoting no direct causal effect of $X$ on $Y$. Independence was determined by Fisher-z tests ($\alpha = 0.001$). All experiments were run on a 2017 MacBook with 2.9 GHz Quad-Core Intel Core i7.

| | | M-Structure | | | | |
|---|---|---|---|---|---|---|
| | Bernoulli \| Linear | | | Hypergeometric \| Quadratic | | |
| $n$ | **Z** Acc | $\mathbf{Z}_1$ Prec | $\mathbf{Z}_1$ Rec | **Z** Acc | $\mathbf{Z}_1$ Prec | $\mathbf{Z}_1$ Rec |
| 500 | 73.5 (71.0-76.0) | 26.2 (17.6-34.9) | 27.0 (18.3-35.7) | 75.3 (73.8-76.8) | 16.0 (8.8-23.2) | 16.0 (8.8-23.2) |
| $1k$ | 92.1 (90.4-93.8) | 90.0 (84.1-95.9) | 90.0 (84.1-95.9) | 87.3 (85.8-88.7) | 94.0 (89.3-98.7) | 94.0 (89.3-98.7) |
| $5k$ | 97.1 (96.0-98.2) | 97.0 (93.6-100) | 97.0 (93.6-100) | 99.8 (99.6-100) | 100 (100-100) | 100 (100-100) |
| $10k$ | 99.7 (99.4-100) | 100 (100-100) | 100 (100-100) | 100 (100-100) | 100 (100-100) | 100 (100-100) |
| | | Butterfly Structure | | | | |
| | Bernoulli \| Linear | | | Hypergeometric \| Quadratic | | |
| $n$ | **Z** Acc | $\mathbf{Z}_1$ Prec | $\mathbf{Z}_1$ Rec | **Z** Acc | $\mathbf{Z}_1$ Prec | $\mathbf{Z}_1$ Rec |
| $1k$ | 60.4 (57.5-63.2) | 16.8 (9.6-24.0) | 12.5 (6.6-18.4) | 61.5 (58.9-64.0) | 28.9 (20.1-37.7) | 16.0 (10.3-21.7) |
| $2.5k$ | 98.8 (97.1-100) | 98.0 (95.2-100) | 98.0 (95.2-100 | 99.9 (99.7-100) | 100 (100-100) | 100 (100-100) |
| $5k$ | 98.9 (97.4-100) | 99.0 (97.0-100) | 98.2 (95.8-100) | 99.9 (99.7-100) | 100 (100-100) | 100 (100-100) |
| $10k$ | 99.7 (99.4-100) | 100 (100-100) | 99.2 (98.4-100) | 99.8 (99.5-100) | 100 (100-100) | 99.5 (98.5-100) |

Table G.6: Performance of Algorithm 1 on 13-node DAGs containing an M-structure structure or butterfly structure (Figure E.1) across noise distributions, causal mechanisms, and sample sizes ($n$). In all DAGs, exposure $X$ is a direct cause of outcome $Y$. Metrics reported are accuracy of all labels (**Z** Acc), mean precision for partition $\mathbf{Z}_1$ ($\mathbf{Z}_1$ Pre), and mean recall for partition $\mathbf{Z}_1$ ($\mathbf{Z}_1$ Rec). The 95% confidence interval is reported in parentheses. Independence was determined by chi-square tests with $\alpha = 0.001$. All experiments were run on a 2017 MacBook with 2.9 GHz Quad-Core Intel Core i7.

| | | Graph with m-structure, butterfly structure, and indirect mediators | | | | |
|---|---|---|---|---|---|---|
| | Bernoulli \| Linear | | | Hypergeometric \| Quadratic | | |
| $n$ | **Z** Acc | $\mathbf{Z}_1$ Prec | $\mathbf{Z}_1$ Rec | **Z** Acc | $\mathbf{Z}_1$ Prec | $\mathbf{Z}_1$ Rec |
| $5k$ | 60.2 (59.0-61.4) | 48.8 (38.9-58.6) | 16.8 (12.4-21.1) | 72.7 (70.2-75.3) | 93.5 (88.7-98.3) | 57.8 (51.4-64.1) |
| $10k$ | 85.8 (82.2-89.4) | 66.5 (57.4-75.6) | 66.2 (57.1-75.4) | 97.9 (96.5-99.2) | 96.9 (93.8-99.9) | 97.0 (94.0-100.0) |
| $15k$ | 97.9 (96.5-99.2) | 96.3 (93.3-99.4) | 96.8 (93.6-99.9) | 98.0 (96.7-99.3) | 96.3 (93.3-99.4) | 97.2 (94.3-100) |
| $20k$ | 98.7 (97.6-99.9) | 97.4 (94.6-100) | 98.0 (95.2-100) | 98.7 (98.0-99.4) | 99.1 (98.1-100.0) | 99.5 (98.8-100) |

Table G.7: Performance of Algorithm 1 on a 17-node DAG featuring an M-structure, butterfly structure, and mediator chain (Figure E.2). Data generating processes represent various discrete noise distributions, linear and nonlinar causal mechanisms, and sample sizes ($n$). Exposure $X$ is a direct cause of outcome $Y$ for all DAGs. Reported values are averaged over 100 DAGs. Metrics reported are mean accuracy of all labels (**Z** Acc), mean precision for partition $\mathbf{Z}_1$ ($\mathbf{Z}_1$ Pre), and mean recall for partition $\mathbf{Z}_1$ ($\mathbf{Z}_1$ Rec). The 95% confidence interval is reported in parentheses. Independence was determined by chi-square tests with $\alpha = 0.005$. All experiments were run on a 2017 MacBook with 2.9 GHz Quad-Core Intel Core i7.

| | COMMON CAUSE CRITERION | | | | | | | | | | | |
|---|---|---|---|---|---|---|---|---|---|---|---|---|
| | VALID ADJUSTMENT SET | | | | CONFOUNDER PRECISION | | | | CONFOUNDER RECALL | | | |
| $n$ | LDP | PC | LDECC | MB-BY-MB | LDP | PC | LDECC | MB-BY-MB | LDP | PC | LDECC | MB-BY-MB |
| 25k | **0.8** | 0.7 | 0.0 | 0.0 | **80.00 (53.87-100)** | 35.00 (20.03-49.97) | 0.0 (0.0-0.0) | 0.0 (0.0-0.0) | **80.00 (53.87-100)** | 35.00 (20.03-49.97) | 0.0 (0.0-0.0) | 0.0 (0.0-0.0) |
| 50k | 0.7 | **1.0** | 0.0 | 0.0 | **76.67 (50.81-100)** | 50.00 (50.00-50.00) | 0.0 (0.0-0.0) | 0.0 (0.0-0.0) | **80.00 (53.87-100)** | 50.00 (50.00-50.00) | 0.0 (0.0-0.0) | 0.0 (0.0-0.0) |
| 75k | **0.9** | 0.4 | 0.0 | 0.0 | **90.00 (80.02-99.98)** | 20.00 (4.00-36.00) | 0.0 (0.0-0.0) | 0.0 (0.0-0.0) | **100 (100-100)** | 20.00 (4.00-36.00) | 0.0 (0.0-0.0) | 0.0 (0.0-0.0) |

| | DISJUNCTIVE CAUSE CRITERION | | | | | | | | | | | |
|---|---|---|---|---|---|---|---|---|---|---|---|---|
| | VALID ADJUSTMENT SET | | | | CONFOUNDER PRECISION | | | | CONFOUNDER RECALL | | | |
| $n$ | LDP | PC | LDECC | MB-BY-MB | LDP | PC | LDECC | MB-BY-MB | LDP | PC | LDECC | MB-BY-MB |
| 25k | 0.8 | **0.9** | 0.3 | 0.9 | **38.00 (25.33-50.67)** | 22.50 (17.60-27.40) | 33.33 (8.03-58.64) | 34.17 (24.91-43.42) | **80.00 (53.87-100)** | 45.00 (35.20-54.80) | 25.00 (8.67-41.33) | 45.00 (35.20-54.80) |
| 50k | 0.7 | **1.0** | 0.2 | 0.7 | **36.33 (23.92-48.75)** | 26.67 (24.49-28.84) | 10.00 (0-23.07) | 23.33 (11.71-34.96) | **80.00 (53.87-100)** | 60.00 (46.93-73.07) | 10.00 (0-23.07) | 35.00 (20.03-49.97) |
| 75k | 0.9 | **1.0** | 0.0 | 0.4 | **45.00 (41.73-48.27)** | 25.83 (24.2-27.47) | 8.33 (0-19.03) | 29.05 (12.20-45.90) | **100 (100-100)** | 50.00 (50.00-50.00) | 12.50 (0-28.54) | 35.71 (17.64-53.79) |

| | BOTH CRITERIA | | | | | | | |
|---|---|---|---|---|---|---|---|---|
| | INDEPENDENCE TESTS | | | | RUNTIME (SECONDS) | | | |
| $n$ | LDP | PC | LDECC | MB-BY-MB | LDP | PC | LDECC | MB-BY-MB |
| 25k | **142.9 (141.5-144.3)** | 3021.9 (2975.2-3068.6) | 2784.1 (2118.4-3449.8) | 823.7 (610.1-1037.3) | **0.065 (0.061-0.069)** | 92.08 (90.351-93.81) | 99.485 (78.085-120.885) | 86.292 (51.39-121.193) |
| 50k | **146.9 (145.2-148.6)** | 3841.9 (3761.2-3922.6) | 4405 (3734.8-5075.2) | 1146.6 (660.5-1632.7) | **0.109 (0.101-0.116)** | 243.973 (237.472-250.474) | 310.255 (262.338-358.172) | 263.322 (145.116-381.528) |
| 75k | **148.6 (147.3-149.9)** | 4307.9 (4225.9-4389.9) | 4615.2 (4049.2-5181.3) | 1567.3 (881.9-2252.7) | **0.162 (0.145-0.178)** | 415.874 (398.714-433.035) | 473.107 (408.198-538.016) | 582.672 (306.342-859.001) |

Table G.8: Baseline comparison on the MILDEW benchmark from `bnlearn` [Scutari, 2010], with MIKRO_1 as exposure and MELDUG_2 as outcome. Independence was determined by chi-square independence tests with $\alpha = 0.005$. Both the common cause criterion and disjunctive cause criterion were considered. Values are reported for 10 replicate DAGs with 95% confidence intervals in parentheses. Sample size is denoted by $n$. Adjustment set quality was measured by fraction that are valid under the backdoor criterion, confounder precision per adjustment set, and confounder recall per adjustment set. The method proposed in this work is highlighted in yellow. The most performant values per metric are bolded. All experiments were run on a 2017 MacBook with 2.9 GHz Quad-Core Intel Core i7. Results are visualized in Figure 6.

| | COMMON CAUSE CRITERION | | | | | | | | | | | |
|---|---|---|---|---|---|---|---|---|---|---|---|---|
| | VALID ADJUSTMENT SET | | | | AVERAGE TREATMENT EFFECT (ATE) | | | | ATE MEAN SQUARED ERROR | | | |
| $n$ | LDP | PC | LDECC | MB-BY-MB | LDP | PC | LDECC | MB-BY-MB | LDP | PC | LDECC | MB-BY-MB |
| 1k | **0.93** | 0.00 | 0.03 | 0.10 | **3.77 (3.75-3.79)** | 3.58 (3.38-3.77) | 3.88 (3.71-4.05) | 3.97 (3.86-4.08) | **0.0096** | 1.0509 | 0.7817 | 0.3703 |
| 2.5k | **0.96** | 0.00 | 0.02 | 0.30 | **3.76 (3.75-3.78)** | 2.5 (2.14-2.87) | 4.08 (4.07-4.09) | 3.97 (3.93-4.01) | **0.0053** | 4.9982 | 0.1088 | 0.0817 |
| 5k | **0.96** | 0.03 | 0.04 | 0.60 | **3.76 (3.75-3.77)** | 1.09 (0.78-1.4) | 4.07 (4.05-4.08) | 3.87 (3.83-3.9) | **0.0046** | 9.5287 | 0.1054 | 0.0473 |
| 7.5k | **0.97** | 0.11 | 0.14 | 0.73 | **3.76 (3.75-3.77)** | 1 (0.72-1.27) | 4.04 (4.01-4.06) | 3.83 (3.8-3.86) | **0.0037** | 9.5009 | 0.0950 | 0.0325 |
| | CONFOUNDER PRECISION | | | | CONFOUNDER RECALL | | | | ADJUSTMENT SET CARDINALITY | | | |
| $n$ | LDP | PC | LDECC | MB-BY-MB | LDP | PC | LDECC | MB-BY-MB | LDP | PC | LDECC | MB-BY-MB |
| 1k | **93 (87.97-98.03)** | 13.17 (8.88-17.45) | 3 (0-6.36) | 10 (4.09-15.91) | **93 (87.97-98.03)** | 27 (18.25-35.75) | 3 (0-6.36) | 10 (4.09-15.91) | **0.9 (0.9-1)** | 0.6 (0.4-0.7) | 0.1 (0-0.1) | 0.1 (0.1-0.2) |
| 2.5k | **96 (92.14-99.86)** | 16.83 (12.34-21.32) | 2 (0-4.76) | 30 (20.97-39.03) | **96 (92.14-99.86)** | 36 (26.54-45.46) | 2 (0-4.76) | 30 (20.97-39.03) | **1 (0.9-1)** | 1 (0.8-1.2) | 0 (0-0) | 0.3 (0.2-0.4) |
| 5k | **96 (92.14-99.86)** | 27.33 (23.19-31.48) | 4 (0.14-7.86) | 60 (50.35-69.65) | **96 (92.14-99.86)** | 70 (60.97-79.03) | 4 (0.14-7.86) | 60 (50.35-69.65) | **1 (0.9-1)** | 2.2 (2-2.4) | 0 (0-0.1) | 0.6 (0.5-0.7) |
| 7.5k | **97 (93.64-100)** | 34.08 (30.81-37.36) | 14 (7.16-20.84) | 73 (64.25-81.75) | **97 (93.64-100)** | 90 (84.09-95.91) | 14 (7.16-20.84) | 73 (64.25-81.75) | **1 (0.9-1)** | 2.7 (2.5-2.9) | 0.1 (0.1-0.2) | 0.7 (0.7-0.8) |
| | DISJUNCTIVE CAUSE CRITERION | | | | | | | | | | | |
| | VALID ADJUSTMENT SET | | | | AVERAGE TREATMENT EFFECT (ATE) | | | | ATE MEAN SQUARED ERROR | | | |
| $n$ | LDP | PC | LDECC | MB-BY-MB | LDP | PC | LDECC | MB-BY-MB | LDP | PC | LDECC | MB-BY-MB |
| 1k | **0.93** | 0.0 | 0.09 | 0.02 | **3.77 (3.76-3.79)** | 0.87 (0.6-1.14) | 1.41 (1.15-1.67) | 1.05 (0.77-1.34) | **0.0091** | 10.2090 | 7.1930 | 9.3487 |
| 2.5k | **0.96** | 0.0 | 0.11 | 0.08 | **3.76 (3.75-3.78)** | 0.28 (0.23-0.33) | 1.31 (1.11-1.51) | 1.34 (1.07-1.61) | **0.0055** | 12.0875 | 6.9627 | 7.6885 |
| 5k | **0.96** | 0.0 | 0.14 | 0.32 | **3.76 (3.75-3.78)** | 0.4 (0.29-0.52) | 2.34 (2.07-2.62) | 2.75 (2.49-3.01) | **0.0050** | 11.5372 | 5.0590 | 3.9600 |
| 7.5k | **0.97** | 0.03 | 0.31 | 0.48 | **3.76 (3.75-3.77)** | 0.6 (0.43-0.77) | 2.52 (2.22-2.81) | 2.75 (2.49-3.01) | **0.0037** | 10.6568 | 3.8045 | 2.7495 |
| | CONFOUNDER PRECISION | | | | CONFOUNDER RECALL | | | | ADJUSTMENT SET CARDINALITY | | | |
| $n$ | LDP | PC | LDECC | MB-BY-MB | LDP | PC | LDECC | MB-BY-MB | LDP | PC | LDECC | MB-BY-MB |
| 1k | **31.83 (29.97-33.7)** | 27.08 (24.77-29.4) | 23.4 (19.08-27.72) | 30.83 (27.95-33.72) | **93 (87.97-98.03)** | 85 (77.97-92.03) | 61 (51.39-70.61) | 87 (80.38-93.62) | **2.8 (2.7-2.9)** | 2.7 (2.5-2.9) | 2.4 (2.2-2.6) | 2.6 (2.3-2.8) |
| 2.5k | **32.33 (30.95-33.71)** | 30.43 (29.51-31.36) | 17.83 (11.86-23.8) | 40.33 (37.88-42.79) | **96 (92.14-99.86)** | 100 (100-100) | 31 (21.89-40.11) | 99 (97.04-100) | **2.9 (2.8-3)** | 3.4 (3.3-3.5) | 1.5 (1.4-1.7) | 2.6 (2.5-2.7) |
| 5k | **32.17 (30.83-33.5)** | 27.5 (26.46-28.54) | 12.5 (7.41-17.59) | 40.92 (38.57-43.26) | **96 (92.14-99.86)** | 99 (97.04-100) | 23 (14.71-31.29) | 98 (95.24-100) | **2.9 (2.8-3)** | 3.7 (3.5-3.8) | 1.2 (1-1.3) | 2.5 (2.4-2.6) |
| 7.5k | **32.5 (31.33-33.67)** | 27.63 (26.49-28.77) | 19.5 (15.05-23.95) | 40.5 (38.56-42.44) | **97 (93.64-100)** | 100 (100-100) | 46 (36.18-55.82) | 99 (97.04-100) | **2.9 (2.9-3)** | 3.8 (3.6-3.9) | 1.5 (1.3-1.7) | 2.6 (2.4-2.7) |

Table G.9: Average treatment effect (ATE) estimation with adjustment sets identified by LDP, PC, LDECC, and MB-by-MB for a 10-node linear-Gaussian DAG (Figure 3). Both the common cause criterion (CCC) and disjunctive cause criterion (DCC) were considered. Values are reported for 100 replicate DAGs with 95% confidence intervals in parentheses. Independence was determined by Fisher-z tests with $\alpha = 0.01$. Adjustment set quality was measured by fraction that are valid under the backdoor criterion, ATE (ground truth = 3.75), ATE mean squared error, confounder precision per adjustment set, confounder recall per adjustment set, and cardinality of the adjustment set (ground truth is 1 under the CCC and 3 under the DCC). The method proposed in this work is highlighted in yellow. The most performant values per metric are bolded. All experiments were run on a 2017 MacBook with 2.9 GHz Quad-Core Intel Core i7. Results are visualized in Figure 6.

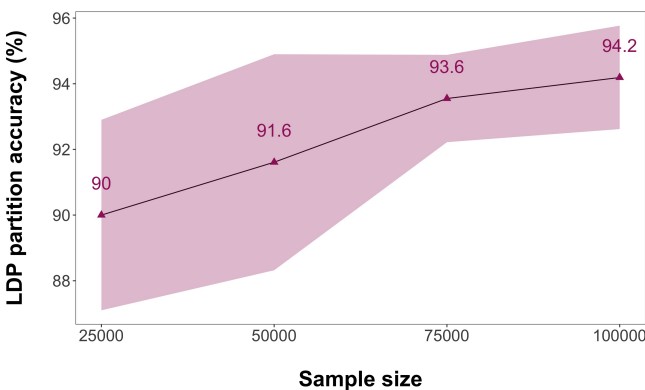

Figure G.2: LDP partition accuracy on the MILDEW benchmark. Mean accuracy was computed for 10 replicate samples from the ground truth DAG using `bnlearn` [Scutari, 2010]. We measure partition accuracy as the percent of partition labels that are consistent with ground truth. Independence was determined by chi-square tests ($\alpha = 0.005$). Shaded regions represent the 95% confidence interval. All experiments were run on a 2017 MacBook with 2.9 GHz Quad-Core Intel Core i7.

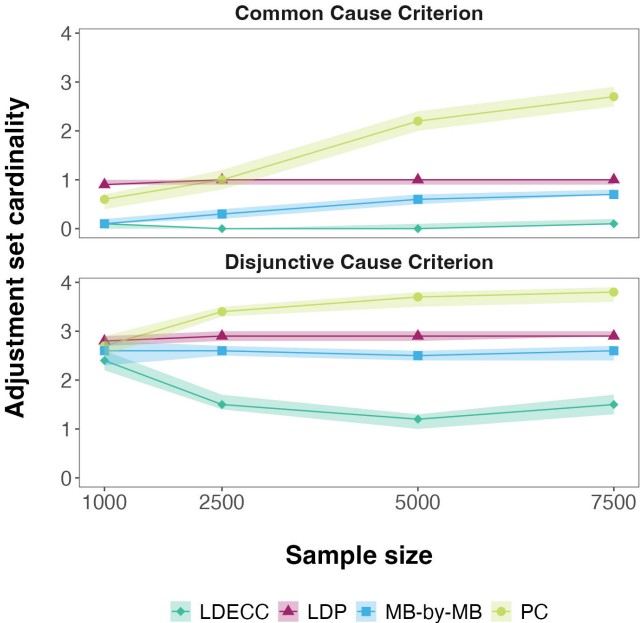

Figure G.3: Adjustment set cardinality for the linear-Gaussian DAG described in Figure 6. The true adjustment set cardinality is one under the common cause criterion and three under the disjunctive cause criterion.

| Latent | VAS Exists | $Z_5$ Crit | % Valid | Z Acc | $Z_1$ Prec | $Z_1$ Rec |
|---|---|---|---|---|---|---|
| $B_1$ | ✓ | ✓ | 100 | 99.00 (98.53-99.47) | 99.50 (98.81-100.0) | 100.0 (100.0-100.0) |
| $B_2$ | ✓ | ✓ | 99 | 99.13 (98.56-99.71) | 98.77 (97.55-99.98) | 99.67 (99.01-100.0) |
| $Z_{4a}$ | ✓ | ✓ | 99 | 98.80 (98.17-99.43) | 99.22 (98.32-100.0) | 99.75 (99.26-100.0) |
| $M_2$ | ✓ | ✓ | 100 | 87.07 (86.46-87.68) | 68.67 (67.11-70.23) | 100.0 (100.0-100.0) |
| $Z_{5a}$ | ✓ | ✓ | 99 | 98.53 (97.63-99.44) | 98.40 (96.34-100.0) | 98.75 (96.73-100.0) |
| $M_1$ | ✓ | ✓ | 100 | 98.87 (98.34-99.39) | 99.47 (98.71-100.0) | 100.0 (100.0-100.0) |
| $Z_1$ | ✗ | ✗ | 0 | 84.40 (82.97-85.83) | 68.92 (64.91-72.92) | 91.67 (86.3-97.03) |
| $B_3$ | ✗ | ✗ | 0 | 73.33 (73.15-73.52) | 41.30 (40.64-41.96) | 67.00 (66.35-67.65) |

Table G.10: Results for numerical validation of Theorem 4.5 on an 18-node ground truth DAG with latent variables (Figures 7, E.5). Values are means over 100 replicates with 95% confidence intervals in parentheses. For each variable dropped from the observed $\mathbf{Z}$ (Latent), we report whether a VAS for the ground truth DAG exists in $\mathbf{Z}$ (VAS Exists), whether the $\mathbf{Z}_5$ criterion passed ($\mathbf{Z}_5$ Crit), the percent of adjustment sets inferred by LDP that were valid with respect to the ground truth DAG (% Valid), partition label accuracy ($\mathbf{Z}$ Acc), precision for partition $\mathbf{Z}_1$ ($\mathbf{Z}_1$ Prec), and recall for partition $\mathbf{Z}_1$ ($\mathbf{Z}_1$ Rec). Causal mechanisms were linear and noise was Bernoulli ($n = 50k$; chi-square tests; $\alpha = 0.001$).

## G.1 IMPACTS OF CONDITIONING SET SIZE

Local baselines faced challenges with chi-square independence tests on MILDEW for $n \geq 75k$. LDECC errored out on 2/10 and 10/10 replicates at $n = 75k$ and $n = 100k$, respectively, while MB-by-MB could not return results for 3/10 and 9/10. Independence test failures persisted even with resampling from the ground truth DAG, and are likely due to large conditioning sets resulting in low or no samples for some groups during binning. While the maximum conditioning set size for LDP on MILDEW was 4, this was 17 for LDECC and 19 for MB-by-MB. Similar sample complexity challenges likely explain our empirical observation that LDP returns VAS for simple discrete DAGs with significantly fewer samples ($n = 1k$) than FCI ($n = 10k$) and PC ($n > 10k$), as the latter methods require many more higher-order independence tests (Figure A.2) [Spirtes et al., 2000].