# OpenReview forum: "Local Discovery by Partitioning: Polynomial-Time Causal Discovery Around Exposure-Outcome Pairs"
_auai.org/UAI/2024/Conference — UAI 2024 poster_

### Official Review · Reviewer_B1Az · 2024-03-12

**Q2-1 Originality-Novelty:** 3
**Q2-2 Correctness-Technical Quality:** 3
**Q2-5 Clarity Of Writing:** 2

**Q1 Summary And Contributions:**

The paper proposes a method to perform local causal discovery by partitioning . The algorithm appears to be polynomial time and has some good characteristics

**Q2-3 Extent To Which Claims Are Supported By Evidence:**

2: Fair: the main claims are somewhat supported by evidence (but the experimental evaluation may be weak, or does not match entirely with the claims, important baselines may be missing, proofs contain important ideas but lack rigor, algorithmic details are only discussed superficially, references are imprecise, assumptions are not sufficiently motivated or explicated, etc.).

**Q2-4 Reproducibility:**

2: Fair: key resources (e.g. proofs, code, data) are unavailable but key details (e.g. proof sketches, experimental setup) are sufficiently well-described for an expert to confidently reproduce the main results.

**Q3 Main Strengths:**

- Polynomial time algorithm
- good motivation
- Appears to be strong contender compared to other methods
- clear limitation and assumptions

**Q4 Main Weakness:**

- There is little to no explanation of the algorithm in the paper, the whole point of the paper is the algorithm and that is relegated to the appendix!

**Q5 Detailed Comments To The Authors:**

as above

**Q9 Complying With Reviewing Instructions:**

Yes

---

> ### Author Rebuttal · Authors · 2024-04-05
>
> We thank the reviewer for their feedback. We agree that space limitations did not result in a self-contained work and appreciate that the reviewer has offered to potentially reconsider their score. We feel that the updates described below will clarify our theoretical contributions and the logic of our algorithm. As the camera-ready for this venue allows for 10 pages, we would no longer face space constraints.
>
> We have linked to images of the new tables and figures (updated text is in red).
>
> **List of items moved from the appendix to the main text:**
> - Figure D.1 (now Figure 4), a schematic of how Algorithm 1 works ([link](https://imgur.com/a/DIu3tAI)).
> - An updated and clarified version of the text from Appendix D.1, which verbally describes each step in Algorithm 1 in plain English ([link](https://imgur.com/a/DIu3tAI)).
> - Tables B.1 and B.2 (now Tables 2 and 3), which describe all possible indirect active path types and the indirect active path combinations that formally define each partition, respectively ([link](https://imgur.com/a/qGepDPk)).
> - Proof of Theorem 3.1,  which states that the causal partition taxonomy is exhaustive and mutually exclusive. This proof was previously in Appendix B.1 and has been updated for clarity ([link](https://imgur.com/a/25rNHiJ)).
> - Table G.10 (now Table 4): partition accuracy, Z1 precision, and Z1 recall under latent confounding ([link](https://imgur.com/a/xCHg4QJ)).
>
> **Formatting changes:**
> To further highlight our empirical results under causal insufficiency, we made the text titled *VAS Discovery With Latent Variables* its own subsection (Sec 6.3).
>
> **New theoretical results to further clarify the algorithm’s behavior with respect to latent confounding and inter-partition active paths.** Note that references to tables and figures use the updated numbering. ([link](https://imgur.com/a/vNVjGjK))
>
> - "*Remark 4.8 (Weakening Causal Sufficiency for Partition Labeling).* We empirically demonstrate in Table 4 that certain forms of latent confounding do not impact LDP’s ability to accurately partition, while others do. As shown in Theorem 4.9, certain tests are robust to causal insufficiency. Thus, it is known that C4 is a sufficient but not necessary condition for correct partitioning. However, we leave the nontrivial task of fully characterizing partitioning behavior under arbitrary latent confounding to future work."
> - "*Theorem 4.9. Steps 1 and 2 of Algorithm 1 are robust to causal insufficiency and inter-partition active paths.* Thus, the identifiability of partitions $\mathbf{Z}_4$ and $\mathbf{Z}_8$ is not impacted by violations of C1 and C4. *Intuition.* Theorem 4.9 follows from the fact that tests for $\mathbf{Z}_4$ and $\mathbf{Z}_8$ rely only on knowledge of {X,Y} and candidate Z. This is in contrast to Step 5, for example, which relies on access to additional variables in the true graph for correctness. Full proof is provided in Appendix D.2."
> - "*Proof of Theorem 4.9.* As shown in Lemmas D.1 and D.2, the following bidirectional statements define $\mathbf{Z}_8$ and $\mathbf{Z}_4$...These statements are unique fundamental properties of $\mathbf{Z}_8$ and $\mathbf{Z}_4$, irrespective of the inter-partition active paths in which they participate. These statements also precisely define the statistical tests used to detect these partitions. Further, these tests rely only on knowledge of $\{X,Y,Z\}$, and no other variables. Thus, failure to observe variables other than $Z$ in the ground truth graph have no impact on the outcome of these tests. This is in contrast to Step 5, for example, which relies on access to additional variables in the true graph for correctness. Thus, Steps 1 and 2 are robust to arbitrary latent confounding and inter-partition active paths."

---

### Official Review · Reviewer_Q8QH · 2024-03-22

**Q2-1 Originality-Novelty:** 3
**Q2-2 Correctness-Technical Quality:** 3
**Q2-5 Clarity Of Writing:** 3

**Q1 Summary And Contributions:**

For purposes such as finding a valid adjustment set, instead of doing full causal discovery, the authors argue that it suffices to know for each covariate what its causal relation is to X and to Y. An algorithm is proposed that partitions the covariates into the eight possible classes, using far fewer independence tests (and with smaller conditioning sets) than required for full causal discovery, and using relatively weak assumptions.

**Q2-3 Extent To Which Claims Are Supported By Evidence:**

2: Fair: the main claims are somewhat supported by evidence (but the experimental evaluation may be weak, or does not match entirely with the claims, important baselines may be missing, proofs contain important ideas but lack rigor, algorithmic details are only discussed superficially, references are imprecise, assumptions are not sufficiently motivated or explicated, etc.).

**Q2-4 Reproducibility:**

3: Good: key resources (e.g. proofs, code, data) are available and key details (e.g. proofs, experimental setup) are sufficiently well-described for competent researchers to confidently reproduce the main results.

**Q3 Main Strengths:**

The core idea, of determining a partition instead of doing full causal discovery, is a good one and would hopefully be applicable in many settings where adjustment sets need to be determined.

The main paper is well-written and appears technically sound (though this is an optimistic estimate that I can't be confident about: see below).

**Q4 Main Weakness:**

The main weakness of this paper is that in my opinion, too much of its contents is in the supplement. The definitions of the different classes of the partition are only given in the main paper for simplified examples, where all variables are related to X or Y by a single directed edge. Latent confounders are not addressed at all in the main paper. The actual definitions are in Appendix B.2 and B.3. This means that most of the theoretical results proving the correctness of the algorithm can't be understood from the main paper alone. Also for many of the cases that are tested in the experimental results, their significance can't be gauged without knowing the full definitions.

**Q5 Detailed Comments To The Authors:**

Questions and concerns:
- In some graphs, adjusting for one variable may open up new paths, so that conceivably, the relation of some other variable to X or Y changes. For the proposed partition categories, is it possible that nodes should be reclassified in such situations?
- I find Definition 3.2 unclear. What is the context here? If the path under consideration is present in the graph, then it should be taken into account when classifying nodes into the partition, so it would seem that the path "complies with" Table 1 by definition of the classification. Or is this about a path not present in the graph at the time of labelling, but added later? If different versions of graphs are considered, this isn't clear from the notation. (Related: in assumption C1, it seems "that are not fully mediated by {X,Y}" is redundant because this condition is already part of the definition of inter-partition active paths.)
- In the introduction, it is suggested that constraint-based methods have the advantage over score-based methods of not requiring parametric assumptions. This is debatable: independence tests make their own (sometimes implicit) assumptions on the forms of the causal mechanisms, so this claim would require better justification. In particular, the experiments with continuous data use an independence test that assumes linear mechanisms, so I don't think it can be said of this paper that it "avoid[s] such assumptions".
- "Empirical results on a 10-node DAG indicate sub-quadratic runtimes (Figure 4)": I don't see how this can be concluded this from the figure: the plots might be quadratic but with constant factor less than 1.
- "Existing nonparametric global methods cannot infer beyond the MEC, which can result in ambiguous partition labeling.": This suggests a nonexistent advantage of LDP over global methods: under the same assumptions C2 and C3 as LDP needs, global methods *can* infer the partition labelling.

Other remarks:
- Throughout: phrases such as "total independence tests" should include "number of".
- For statements such as in the caption of Figure 1 and elsewhere: I don't think you wrote anywhere what $n$ and $k$ are.
- In Definition 2.5 ("Confounder"), can Z be latent? Can elements of S be latent?
- Algorithm 1, line 8: what is $Z_{5A,7}$? It doesn't appear elsewhere in the algorithm. Should it be $Z_{5,7}$?
- Caption Figure E.1: "right" and "left" switched around
- There is a blank page at the end of the supplement.


---

**EDIT in response to rebuttal:**

> "Thank you for the clarifying question. We have added the following simple DAG as a counterexample in our updated appendix (link). ..."

This example is incorrect. The DAG in which the three nodes Z1abc form an M-structure is *not* consistent with the output of PC, as it adds a v-structure that should not be there according to the output of PC. Indeed, what you are claiming here is by definition impossible: Your method, which uses just conditional independence tests, can't get more information about the causal graph than could be learned from the Markov equivalence class. So please remove from your paper the suggestion that LDP has an advantage here over global causal discovery methods.

Other things I encountered while writing this reply:

* Definition 4.3 suggests the condition there should be the same as in line 38 of the algorithm, but the conditions are different;

* Caption of Table B.1: "Types 1 and Type 2" should be "Type 2 and Type 3";

* Table B.1, the two "Type 4" descriptions: should these exclude the case that the path passes through Y resp. X?

**Q9 Complying With Reviewing Instructions:**

Yes

---

> ### Author Rebuttal · Authors · 2024-04-05
>
> We thank the reviewer for their feedback. We agree that space limitations did not result in a self-contained work. We feel that our revisions, including moving Appendices B.1 and D.1 to the main text, have clarified our theoretical contributions and the logic of our algorithm. Please refer to the rebuttal to Reviewer B1Az for full details on changes made.
>
> We have addressed all minor comments in the revision. Next, we will address your major comments.
>
> > *"Latent confounders are not addressed at all in the main paper."*
>
> In the original submission, robustness to causal insufficiency for valid adjustment set (VAS) discovery is noted in the abstract and introduction (*Organization*), and is discussed at length in Sec 4.1 (*Sufficient Conditions for VAS Identification*). Sec 6 (*VAS Discovery With Latent Variables*) provides empirical validation. For partition correctness, we provide empirical evidence that certain forms of inter-partition paths and latent confounding do not impact correctness (Sec 6, *Partition Label Correctness*, Table G.10 [[now Table 4]](https://imgur.com/a/xCHg4QJ)). To further clarify, we have added a new theoretical result (Thm 4.9) that further characterizes partitioning behavior in the presence of inter-partition active paths and latent confounding ([link](https://imgur.com/a/vNVjGjK)).
>
> > *"The actual definitions are in Appendix B.2 and B.3."*
>
> These have been moved to the main text ([link](https://imgur.com/a/qGepDPk)).
>
> > *"In some graphs, adjusting for one variable may open up new paths..."*
>
> This is a valid clarification question. Partitions are defined with respect to the ground truth DAG, not the observed graph. Adjustment in the observational setting is a statistical technique that does not alter the ground truth graph. Thus, the relationship of a variable to $X$ and $Y$ does not change with respect to the conditioning set. We have updated the language to emphasize this (bold added): *"Rather, they are universal properties of any **ground truth** DAG with respect to a chosen exposure-outcome pair."*
>
> > *"I find Definition 3.2 unclear..."*
>
> With the definition of inter-partition active paths, we are defining a kind of path that may or may not exist in the true graph. Different versions of the graph are not considered and there is no time series component to this work. We have moved Appendix B.1 to the main text and now reference what was formerly Table B.2 in the definition for improved clarity ([link](https://imgur.com/a/qGepDPk)). Re compliance with partition definitions: In our original submission, we provide an example "non-permissible inter-partition active path" that cannot exist (just below Def 3.2).
>
> > *In the introduction, it is suggested that constraint-based methods have the advantage..."*
>
> We apologize for the confusion. We mean that constraint-based methods do not innately require parametric assumptions for the identification of the causal DAG. In contrast, score-based methods often require the additive noise assumption, which is not identifiable for all parametric forms (e.g., SCORE, DAS; Montagna et al. 2023). Constraint-based methods rely on the output of independence tests for the identification of the causal DAG. These independence tests can be carried out either parametrically (e.g., Fisher z) or nonparametrically (e.g., KCI, CMI). To clarify, we have revised the first sentence in the second paragraph of the intro to *"We consider the set of causal discovery methods that do not require parametric assumptions on the underlying data generating process for identifiability of causal relations, ..."* We have also added to Sec 4.1: *"While user-specified independence tests might impose their own parametric assumptions, nonparametric tests are recommended when the data generating process is unknown (e.g., Zhang et al. 2011, Runge 2018b)."* In our experiments, we sometimes use parametric tests only for ease of exposition on known data generating processes.
>
> > *"Empirical results on a 10-node DAG indicate sub-quadratic runtimes..."*
>
> We agree and have removed this claim.
>
> > *"Existing nonparametric global methods cannot infer beyond the MEC, which can result in ambiguous partition labeling..."*
>
> Thank you for the clarifying question. We have added the following simple DAG as a counterexample in our updated appendix ([link](https://imgur.com/a/Jym0Eav)). Even in the presence of a $\mathbf{Z}4$ and $\mathbf{Z}5$ (satisfying C2 and C3), PC with an oracle is unable to produce unambiguous causal partitions. Interpretations consistent with the returned MEC are that these three nodes form an M-structure (where $Z{1a} \in \mathbf{Z}2$, $Z{1b} \in \mathbf{Z}5$, and $Z{1c} \in \mathbf{Z}4$) or that they form a backdoor path where all nodes are in $\mathbf{Z}1$. Differentiating M-structures from backdoor paths is a canonical problem in covariate selection. In contrast, LDP correctly labeled $Z1a$, $Z1b$, and $Z1c$ as confounders.

---

### Official Review · Reviewer_udWY · 2024-03-22

**Q2-1 Originality-Novelty:** 3
**Q2-2 Correctness-Technical Quality:** 3
**Q2-5 Clarity Of Writing:** 2

**Q1 Summary And Contributions:**

After rebuttal, score increased from 4 to 5.
---
The authors propose a polynomial-time local causal discovery method that works by partitioning the variables around an exposure-outcome pair into different subsets based on their relation to the exposure-outcome pair. Under the assumption of causal sufficiency, the authors distinguish the eight possible causal models showcasing the relationship between the exposure, the outcome, and a third variable. Assuming that these *partitions* are not connected by active paths, they describe an algorithm for *local discovery by partitioning* (LDP) which is designed to identify these partitions. In the experimental section, the authors show how well LDP can identify these partitions to ultimately identify quality adjustment sets, corresponding to one of the defined partitions, that can be used in downstream causal effect estimation tasks.

**Q2-3 Extent To Which Claims Are Supported By Evidence:**

3: Good: the main claims are supported by convincing evidence (in the form of adequate experimental evaluation, proofs, (pseudo-)code, references, assumptions).

**Q2-4 Reproducibility:**

2: Fair: key resources (e.g. proofs, code, data) are unavailable but key details (e.g. proof sketches, experimental setup) are sufficiently well-described for an expert to confidently reproduce the main results.

**Q3 Main Strengths:**

The authors have come with a rather interesting approach for local causal discovery, from which I like the focus on downstream tasks like causal effect estimation. The experimental section is well-designed, including an independent data set and a sensitivity analysis, and showcases convincing results, especially regarding the number of independence tests performed. The methodology also seems sound, but only after reading the appendix.

**Q4 Main Weakness:**

The authors have clearly put a lot of work into this approach, but the presentation in the main paper is lacking, as crucial details are deferred to the appendix (for example, in D.1 and B.2). Before reading the appendix, it was not clear to me how the algorithm is supposed to work and I was not sure it was entirely sound. Highlighting the length 1 paths in Figure 2 is potentially misleading since there are steps in the algorithm (e.g., Step 5) that require longer intra-partition paths in order to be triggered.

A second point of contention for me is that the assumptions made are potentially very strong, especially condition $C1$, which forbids any active paths between the different partitions. Even if some of the active paths are not permissible, I am not sure how reliable it would be to assume for example that post-treatment variables $\boldsymbol{Z_4}$ and $\boldsymbol{Z_6}$ are independent from each other. Additionally, causal sufficiency is typically a rather restrictive assumption in a real-world setting.

**Q5 Detailed Comments To The Authors:**

- In the abstract, do you mean "**The total number of independence tests is in the worst-case quadratic in the number of variables.**"?
- On page 1, right column, first full paragraph: "score-based methods" are not introduced or explained anywhere in the paper.
- On page 1, you claim that "nonparametric methods require strong assumptions on the ordering of variable variances", but that does not hold for constraint-based methods like PC/FCI, which are also nonparametric. You also say as much in Section 6 (Baseline Discovery Methods). Perhaps you were referring to a specific method, but the claim seems too general. Could you elaborate on this?
- On page 1, you claim that your LDP approach "does not assume pretreatment", which sounds rather vague. Do you mean to say "**does not assume that all variables outside the exposure-outcome pair are pre-treatment**"?
- In Definitions 2.1 and 2.2. you refer to undirected paths, yet the active paths that you describe appear in a directed graph and contain colliders, so they are clearly directed.
- In Section 3 (first paragraph) you say "partitions are not assumptions on the true DAG", which is true, but hides the fact that the DAG itself implies a certain data generating process, encoding assumptions such as acyclicity and causal sufficiency
- In Section 3 (before Definition 3.2) you describe subpartitions that are "descended from or adjacent to a specific variable". It would have been very useful to show what these subpartitions look like. For example, it is not immediately clear what $\boldsymbol{Z_{2 \in de(X)}}$ would be just by looking at Figure 2, so one has to consider paths of length larger than one.
- Algorithm 1: In line 8, is $\boldsymbol{Z_{5A,7}}$ different from $\boldsymbol{Z_{5,7}}$? If so, what is the difference? In step 5, how can $\boldsymbol{Z_{1,2,3}} \subset \boldsymbol{Z_{\textrm{MIX}}}$ be separated from $Y$ since they are adjacent to $Y$? In step 6, why define a variable $Z_{\textrm{MIX}}$ that does not necessarily belong in the subset of the same name ($\boldsymbol{Z_{\textrm{MIX}}}$, line 22)? Is $Z_{\textrm{MIX}}$ any variable in $\boldsymbol{Z'}$? In line 27, why is $Z_{\textrm{MIX}} \in \boldsymbol{Z_1} \textrm{, but } \notin \boldsymbol{Z_5}$? Wouldn't two nodes in $\boldsymbol{Z_5}$ also be marginally independent? In general, it would be very helpful to clearly specify the subset each iterated variable comes from, even if redundant.
- On page 5, after presenting the "Sufficient Conditions for Correct Partitioning" you mention that condition "C2 in turn guarantees that all backdoor paths will be blocked by the conditioning set in Step 5 of Algorithm 1, which is used to discover $\boldsymbol{Z_5}$". Are you referring to $\boldsymbol{Z_{\textrm{MIX}}}$ here? From the definition in Step 5 and from D.1, it does not seem that this set would be enough to block all backdoor paths between $X$ and $Y$. For example, what if $Z_1$ is a single covariate as depicted in Figure 3? In that case, it does not seem like it would be included to $\boldsymbol{Z_{\textrm{MIX}}}$ in Step 5, since it cannot be separated from $Y$.
- I do not think the claim in Remark 4.7 that "assuming the presence of at least one verifiable representative from a single partition ($\boldsymbol{Z_4}$) is more moderate than assuming the complete absence of multiple partitions" is so easily quantifiable. How can we be reasonably certain about the existence of proxy variables $\boldsymbol{Z_4}$ that are only associated with $Y$?
- In the experimental section, why did you only run 5 replicates for FCI? Are you only showing one of the replicates in Figure 7? It seems rather peculiar that FCI cannot even pick up the marginal dependency between $X$ and $Z_1$ with 50 thousand samples. How do you explain this result? What was the correlation etween $X$ and $Z_1$ and which test rendered these two independent?

**Q9 Complying With Reviewing Instructions:**

Yes

---

> ### Author Rebuttal · Authors · 2024-04-05
>
> We thank the reviewer for their feedback. We agree that space limitations did not result in a self-contained work. To address this, we have moved many details from appendix to main body and provided new theoretical results. Please refer to our rebuttal for Reviewer B1Az for details.
>
> We have addressed all minor comments in the revised text. We address major comments below.
>
> **Causal insufficiency and inter-partition active paths (IPAP):** We would like to clarify that Conditions C1 and C4 are not required for the correctness of VAS discovery, as stated in Thm 4.5. Robustness to causal insufficiency for VAS discovery is noted in the abstract and introduction (*Organization*), and is discussed at length in Sec 4.1 (*Sufficient Conditions for VAS Identification*). Sec 6 (*VAS Discovery With Latent Variables*) provides empirical validation. For partition correctness, we provide empirical evidence that certain forms of IPAP and latent confounding do not impact correctness (Sec 6, *Partition Label Correctness*; Table G.10 [[now Table 4]](https://imgur.com/a/xCHg4QJ)). New theoretical results (Thm 4.9) further characterize partitioning behavior under IPAP and latent confounding [link](https://imgur.com/a/vNVjGjK).
>
> > "On page 1, you claim..."
>
> We have clarified that we mean nonparametric *score-based* methods.
>
> > *"In Section 3 (first paragraph)..."*
>
> The causal partitions are properties wrt the true graph irrespective of their identifiability from observed data. The quoted text is explicitly worded wrt DAGs, hence acyclicity is not hidden. When discussing fundamental properties of the true graph, there is no notion of causal (in)sufficiency. This concept only becomes salient in the research setting, when some variables may be inaccessible. We declare all identifiability conditions in Sec 4.1.
>
> > "In Definitions 2.1 and 2.2..."
>
> This follows a standard definition and does not imply that the paths are undirected, but rather that you don’t account for directionality. We have added the following footnote: "Note that these definitions consider both the directed path and its corresponding undirected path, ignoring directionality."
>
> > "In Section 3 (before Definition 3.2)..."
>
> We have added reference to Fig E.1, where counterexample $M_3$ is in $\mathbf{Z}_2$ but is not adjacent to $Y$ nor descended from $X$.
>
> > Confusion about $\mathbf{Z}_{mix}$.
>
> $Z{mix} \in \mathbf{Z}{mix}$ is not a variable in $Z'$. This superset is described in Sec 4 and p17 of the appendix. The latter has been moved to the main text ([link](https://imgur.com/a/DIu3tAI)).
>
> > *In step 5, how can $\mathbf{Z}{1,2,3}$ be separated from $Y$ since they are adjacent to $Y$?*
>
> $\mathbf{Z}_1$, $\mathbf{Z}_2$, and $\mathbf{Z}_3$ can lie along arbitrarily long active paths to $X$ and $Y$. Thus, some members of these partitions are not adjacent to $Y$ (as is true with confounders, colliders, and mediators as classically defined; e.g., $B_1 \in \mathbf{Z}_1$, Figure E.1).
>
> > *"In line 27..."*
>
> We have clarified this in the revised text ([link](https://imgur.com/a/DIu3tAI)).
>
> > *"On page 5..."*
>
> We have updated to clarify: “blocked by the conditioning set ($X \cup \mathbf{Z}' \setminus Z$) in Step 5...”
>
> > *"I do not think the claim in Remark 4.7..."*
>
> The test for $\mathbf{Z}_4$ relies only on {X,Y,Z}. Low-order conditional independence tests are relatively reliable statistical tests (see *Sample Complexity*). Our new theoretical results also show that the test for $\mathbf{Z}_4$ is robust to IPAP and causal insufficiency ([link](https://imgur.com/a/vNVjGjK)). Verifying that all variables are pretreatment is a signifcantly harder statistical task with no well agreed upon procedure; hence, it is traditionally an assumption. Further, it is simpler to apply expert knowledge to one variable than to many.
>
> > *FCI behavior.*
>
> The unexpected results in Fig 7 and Fig 1 are central to the argument of this paper: global nonparametric discovery displays failure modes for causal partitioning in the finite sample regime (intro; Sec 5]). We have updated the text to clarify that Fig 7 follows the same experimental setup as the paragraph in which it was described, and the graph is consistent with all 5 replicates.
>
> We ran 5 replicates because the interpretation of FCI for VAS discovery is nontrivial, due to bidirected edges and uncertain directionality (open circles) introducing ambiguity (which we further demonstrate in the oracle setting here [[link]](https://imgur.com/a/Jym0Eav)). Thus, we manually inspected five replicates. We have clarified this point in our revised paper.
>
> We assessed stability across random seeds and sensitivity to the significance level of the independence test ($\alpha \in [0.05..0.001]$). Correlation coefficients are 0.26 (95\% CI [0.25, 0.27]) for (X,Z1) and 0.26 ([0.25, 0.27]) for (Y,Z1). Under no setting did FCI produce a VAS. We hypothesize that poor sample complexity wrt conditioning set size is explanatory (see Sec 4 *Sample Complexity*).

---

### Official Review · Reviewer_CEjM · 2024-03-22

**Q2-1 Originality-Novelty:** 3
**Q2-2 Correctness-Technical Quality:** 3
**Q2-5 Clarity Of Writing:** 4

**Q1 Summary And Contributions:**

In their paper the authors propose a local (requiring knowledge and existence of a particular cause-effect pair) nonparametric causal discovery procedure that is based on partitioning given variables into (exhaustive) partition categories.

**Q2-3 Extent To Which Claims Are Supported By Evidence:**

3: Good: the main claims are supported by convincing evidence (in the form of adequate experimental evaluation, proofs, (pseudo-)code, references, assumptions).

**Q2-4 Reproducibility:**

4: Excellent: key resources (e.g. proofs, code, data) are available and key details (e.g. proof sketches, experimental setup) are comprehensively described for competent researchers to confidently and easily reproduce the main results.

**Q3 Main Strengths:**

Given a cause-effect pair finding valid adjustment sets is an important task in causal inference and can be solved by first learning the global causal structure, a notoriously hard problem. Unfortunately, cause-effect based local discovery lies somewhat outside my field of expertise and I can't really say much more than that I enjoyed the paper. It is convincing, well-written, and technically sound.

**Q4 Main Weakness:**

See above.

**Q5 Detailed Comments To The Authors:**

Nothing to add here.

**Q9 Complying With Reviewing Instructions:**

Yes

---

> ### Author Rebuttal · Authors · 2024-04-05
>
> We thank the reviewer for their time in providing this feedback, and for their recognition of our contributions.

---

### Official Review · Reviewer_uhGU · 2024-03-22

**Q2-1 Originality-Novelty:** 3
**Q2-2 Correctness-Technical Quality:** 3
**Q2-5 Clarity Of Writing:** 4

**Q10 Ethical Concerns:**

No.

**Q1 Summary And Contributions:**

The paper considers the problem of causal discovery with respect to a given exposure-outcome pair. More specifically, the paper proposes a constraint-based non-parametric local approach where the goal is to partition the considered variables into subsets defined by their causal relations to the exposure-outcome pair. The identified partitions can then be used in subsequent causal inference tasks, for example, to adjust for confounding bias, avoididing the pretreatment assumption. Sufficient conditions for identifying the correct partitioning and, importantly, a valid adjustment set are given. The asymptotic guarantees and the overall utility of the proposed method are evaluated and demonstrated in numerical experiments.

**Q2-3 Extent To Which Claims Are Supported By Evidence:**

4: Excellent: all claims are supported by very convincing evidence (in the form of comprehensive experimental evaluation, rigorous mathematical proofs, detailed (pseudo-)code, precise references, well-motivated and realistic assumptions) and the authors deliver what they promise.

**Q2-4 Reproducibility:**

4: Excellent: key resources (e.g. proofs, code, data) are available and key details (e.g. proof sketches, experimental setup) are comprehensively described for competent researchers to confidently and easily reproduce the main results.

**Q3 Main Strengths:**

A novel, interesting and useful idea. Convincing proofs and empirical evidence. Very well-structured and -written paper.

**Q4 Main Weakness:**

No main weakness.

**Q5 Detailed Comments To The Authors:**

In terms of the numerical experiments, it would be interesting to see similar comparisons for a few other networks from the bnlearn repository, in addition to the Mildew network. The Mildew network is a bit special in the sense that is contains several high-cardinality variables, which I imagine can be a bit challenging when performing CI tests. This is perhaps the reason behind the seemingly large sample sizes in Fig 6A-E(?)

**Q9 Complying With Reviewing Instructions:**

Yes

---

> ### Author Rebuttal · Authors · 2024-04-05
>
> We thank the reviewer for their time in reviewing our paper, and for their recognition of the novelty and contributions of this work.
>
> *With respect to additional bnlearn benchmarks:* We appreciate the insight that the high-cardinality variables present in Mildew make this benchmark relatively hard to learn with independence tests. We agree that this likely contributes to the large sample sizes required for all baselines. Given space limitations, we have opted to defer further experiments to future works. However, we thank the reviewer for this suggestion and will consider this moving forward.

---

### Meta-Review · Area_Chair_5X1B · 2024-04-18

The paper presents a new efficient algorithm for local causal discovery aimed at finding/identifying valid adjustment sets (VAS) for causal effect estimation of a target pair of exposure-outcome variables X ->Y, subject to a number of assumptions on the local causal structure. It is based on partitioning related variables into one of several disjoint categories based on their causal relation to the pair (X,Y), which can subsequently be used to provide the required VAS. Experiments show the method is faster and obtains more accurate / less biased adjustment sets than baseline (global) discovery algorithms.

Main strengths mentioned by reviewers:
- important and challenging problem
- novel and effective approach
- well-written paper

Weaknesses:
- strong assumptions that may be unlikely to be met in practice
- not sufficiently self-contained: crucial details needed to understand the method and its limitations are deferred to the appendix

Reviewer scores were initially borderline to positive with some concerns about soundness, but after extensive author rebuttal and a helpful discussion among reviewers tended to converge on accept.

Having read the paper I agree with the reviewers that it is a solid contribution and an interesting approach, provided the authors try to incorporate some of the key details from the supplement in the main paper (see reviewer comments).
Having said that, I also have to admit I am more critical of the actual significance of the approach presented. The core conditions  C1-C4 (p.5) behind the method are actually quite restrictive, and could potentially lead to misinterpretations (i.e. wrong/missed causal conclusions) for researchers that are unaware of the impact of these, when not satisfied.
Furthermore, the claimed/obtained increase in speed and accuracy of the method over other, global approaches (like running PC and then finding valid adjustment sets), seems largely predicated on these assumptions as well. Effectively it turns the general problem into a much simpler version, and indeed if we can assume C1-C4 hold, then the proposed method is very fast and effective. In other words: a (large) part of the performance improvement is by assuming away the problem. But there exist plenty cases where PC can find a perfect adjustment set but the current method will fail as e.g. C2 or C3 are not satisfied.

However, at least the paper is explicit in stating these underlying assumptions, and when they do apply the method is indeed sound and effective. And I think the partitioning idea itself is actually quite interesting and could likely lead to other applications as well. Therefore I will follow the reviewer consensus and recommend (borderline) accept, albeit with one important caveat, below.

CAVEAT:
If accepted in the conference, the authors need to correct the error in their counter-example in answer to reviewer Q8QH:

--- excerpt from reviewer discussion ---
Q8QH: Their response to my final remark is clearly wrong:

Author quote: "Thank you for the clarifying question. We have added the following simple DAG as a counterexample in our updated appendix (link). ..."

Q8QH: What the authors are claiming here is that their method, using just conditional independence tests, can get more information than could be learned from the Markov equivalence class. This is by definition impossible, and indeed the example is incorrect. The DAG in which the three nodes Z1abc form an M-structure is not consistent with the output of PC, as it adds a v-structure that should not be there according to the output of PC.
--- end of excerpt ---

The explanation in linked Figure A.3 is indeed wrong: the PC-MEC is *not* ambiguous, and LDP *cannot* find information beyond the MEC using only independence tests. In particular, the CPDAG from PC does *not* imply the presence of an M-structure (only the presence of an active backdoor path), and even though LDP may correctly identify Z_1a, Z_1b, and Z_1c as valid adjustment sets (as does PC), it is not correct to label all three of them as identifiable 'confounders' of X->Y in the MEC shown.